

# C-Coupler2: a flexible and user-friendly community coupler for model coupling and nesting

Li Liu[1,2], Cheng Zhang[1,2], Ruizhe Li[1,2], Bin Wang[1,2,3]

[1]Ministry of Education Key Laboratory for Earth System Modeling, Center for Earth System Science (CESS), Tsinghua

University, Beijing, China

[2]Joint Center for Global Change Studies (JCGCS), Beijing, China

[3]State Key Laboratory of Numerical Modeling for Atmospheric Sciences and Geophysical Fluid Dynamics (LASG), Institute of Atmospheric Physics, Chinese Academy of Sciences, Beijing, China

*Correspondence to:* L. Liu (liuli-cess@tsinghua.edu.cn), Cheng Zhang (zhangc-cess@tsinghua.edu.cn)

**Abstract.** The Chinese C-Coupler (Community Coupler) coupler family aims primarily to develop coupled models for weather forecasting and climate simulation and prediction. It is targeted to serve various coupled models with flexibility, user-friendliness, and extensive coupling functions. C-Coupler2, the latest version, includes a series of new features in addition to those of C-Coupler1, including a common, flexible, and user-friendly coupling configuration interface that combines a set of application programming interfaces and a set of XML formatted configuration files, capability of model

coupling within one executable or the same subset of MPI (Message Passing Interface) processes, flexible and automatic coupling generations that enable coupling generation at any time for any subset of component models, dynamic 3-D coupling that enables convenient coupling of fields on 3-D grids with time-evolving vertical coordinate values, non-blocking data transfer, facilitation for model nesting, facilitation for increment coupling, and debugging capability. C-Coupler2 is ready for use to develop various coupled or nested models. It has passed a number of test cases derived from a sample model with

model coupling and nesting, and with various MPI process layouts between component models, and has already been used in several real coupled models.

## 1   Introduction

Couplers are programs that can handle data interpolation and data transfer between different models and different grids. They have been widely used to develop coupled models for fields such as weather forecasting and climate simulation and

prediction. The Community Coupler (C-Coupler) family was initiated in 2010 in China. It aims to flexibly serve various coupled models with a user-friendly interface and to provide extensive coupling functions to complement the ever increasing number of coupled models being developed and used in China. To this end, the first version (C-Coupler1; Liu et al., 2014) included new features such as flexible coupling configuration based on configuration files and 3-D coupling capability. There are two representative applications of C-Coupler1. The first is a coupled climate system model version FGOALS-gc





(Liu et al., 2014) that was built by replacing the original coupler CPL6 (Craig et al., 2005) used in the CMIP5 (Coupled Model Intercomparison Project, phase 5) model FGOALS-g2 (Li et al., 2013a) by C-Coupler1. FGOALS-gc can achieve exactly the same (bitwise identical) simulation results as FGOALS-g2, and was further used in several Chinese research projects. The second application is a regional coupled model FIO-AOW (Zhao et al., 2017) that consists of an atmosphere

model WRF (Wang et al., 2014), an ocean model POM (Wang et al., 2010), and a wave model MASNUM (Yang et al., 2005). FIO-AOW employs both 2-D and 3-D coupling, where the coupling from MASNUM to POM includes a 3-D field, the wave-induced mixing coefficient (Qiao et al., 2004). FIO-AOW has been used in research to improve typhoon forecasting (Zhao et al., 2017). These applications demonstrate that C-Coupler1 can be used for various coupling configurations. C-Coupler1 demonstrates the feasibility of the general design of C-Coupler, although as the first version, it

does not fully achieve all our targets and still has some limitations. A significant limitation is that C-Coupler1 is not sufficiently user-friendly. To construct a coupled model with C-Coupler1 requires much effort to prepare the configuration files in ASCII format. For example, there are about 2500 and 700 lines in the configuration files for FGOALS-gc and FIO-AOW, respectively. Considering this, and that the next C-Coupler versions may not be compatible with C-Coupler1, we did not aim to achieve wide usage of C-Coupler1, but sought to overcome the limitations while guaranteeing backwards

compatibility in subsequent C-Coupler versions.

The next version, C-Coupler2, includes not only increased coupling function support but also flexibility and user-friendliness. It is now ready for use and has passed hundreds of test cases based on a sample coupled model and several real coupled models. Compared with C-Coupler1 or even other existing couplers, C-Coupler2 has the following new features.

1) A common, flexible, and user-friendly coupling configuration interface that combines a set of application programming

interfaces (APIs) and a set of configuration files in XML format. This interface enables the user to flexibly and conveniently specify or change coupling configurations including the component models used in a coupled model, the time step of each component model, model grids, parallel decompositions on model grids, coupling frequencies, coupling lags between component models, the fields to be coupled, the data type of coupling fields, coupling connections between component models, and coupling generations. Remapping configurations are also modifiable: the

user can either use the remapping weights that are read from an existing remapping weight file produced by software tools such as SCRIP (Jones, 1999), ESMF (Hill et al., 2004), YAC (Hanke et al., 2016), and CoR (Liu et al., 2013), or use remapping weights that are automatically generated by C-Coupler2 in parallel.

2) Model coupling within one executable or the same subset of MPI (Message Passing Interface) processes. The component models that are coupled together can be in either multiple executables or in one, and can fully or partially

share the same subset of MPI processes. Different processes in the same component model can also be coupled with C-Coupler2.

3) Flexible and automatic coupling generation. The coupling generator can automatically detect existing component models in the coupled model, detect possible coupling connections within a subset of component models, and generate a coupling procedure for each coupling connection. A coupling procedure can include data transfer, data interpolation,



data type transformation, and data averaging when necessary. Multiple coupling generations can be performed for a coupled model, and a coupling generation can be performed at any time for any subset of component models.

4) Dynamic 3-D coupling capability. This allows convenient coupling of fields between two 3-D grids, either of which has variable vertical coordinate values that change in time integration.

5) Non-blocking data transfer. It is implemented with one-sided MPI communication (i.e., MPI_put and MPI_get), to minimize potential deadlocks, achieve effective overlap between data transfer and model computation, and enable flexible setting of a lag in model coupling (given two component models that are coupled, they can have different model times in a coupling exchange).

6) Facilitation for model nesting. C-Coupler2 facilitates a regional model (either a component model or a coupled model constructed with a coupler) to be nested (either one way or two ways) into itself or another model without significant changes to the model codes, and can enable different grid domains in a nested system to be integrated simultaneously for better parallel performance.

7) Facilitation for incremental coupling. An existing coupled model coupled by any coupler can be used as a component model by C-Coupler2, which employs the component models of the existing coupled model as its child components. Thus, an existing coupled model can be coupled with another model (itself either a single component model or a coupled model) to make a bigger coupled model, where only the new model coupling through C-Coupler2 requires to be newly developed, without changes to the original coupling in the existing coupled model.

8) Debugging capability. A series of implementations in C-Coupler2 enable to alert the user, as early as possible, to the potential risks of constructing or using a coupled model, and guide the user to fix the errors in model codes and configuration files.

The remainder of this paper is organized as follows. We briefly review C-Coupler and C-Coupler1 in Section 2, introduce the motivation for the development of C-Coupler2 in Section 3, describe the implementation of C-Coupler2 in Section 4, evaluate C-Coupler2 in Section 5, and discuss and conclude C-Coupler2 in Section 6.

## 2 Brief review of C-Coupler and C-Coupler1

Figure 1 shows the two key aspects of the general architecture of models coupled with C-Coupler. First, C-Coupler can serve different coupling configurations in various coupled models. Second, a model can have an identical code version in coupled models with different coupling configurations. This general architecture can be achieved through C-Coupler's software structure (Fig. 2), which consists of a coupling configuration system, a coupling generator, and a runtime software system. The runtime software system works a common, flexible, and extendable library that includes various coupling functions or can even integrate external coupling algorithms (e.g., flux computation algorithms) to serve various coupling configurations. The configuration system defines common rules for describing various coupling configurations. Besides the coupling configuration of component models and coupled models, the configuration system includes a runtime configuration,





which describes detailed coupling procedures corresponding to a coupling configuration. This is the input of the runtime software system. The coupling generator can automatically generate the runtime configuration, facilitating the construction of a coupled model.

The first version, C-Coupler1, was not developed with the aim of achieving the full software structure in Fig. 2, but with a focus only on the runtime software system and the runtime configuration (Fig. 3). To describe the runtime configuration, a set of ASCII configuration files were designed. Besides the traditional coupling functions of data transfer and data interpolation for 2-D coupling, the runtime software system of C-Coupler1 can integrate external coupling algorithms and has 3-D coupling capability that enables convenient coupling of fields between different 3-D grids. To achieve simultaneous 2-D and 3-D coupling, remapping software CoR1 was developed and included in C-Coupler1. CoR1 can effectively manage 1-D, 2-D, and 3-D grids, and can interpolate the fields on such grids, where the 3-D interpolation is performed in a "2-D + 1-D" manner ("2-D" corresponds to interpolation between horizontal sub-grids, and "1-D" corresponds to interpolation between vertical sub-grids).

## 3    Motivation

We considered the following motivations when designing and developing C-Coupler2.

### 3.1    Coupling configuration

Although the configuration system was not fully developed in C-Coupler1, experience gained from the runtime configuration of C-Coupler1 is valuable when designing the coupling configuration of component models and coupled models. The runtime configuration is almost fully based on configuration files, which can improve flexibility in specifying or changing the coupling configuration, but their overuse may significantly lower user-friendliness. For example, many changes to configuration files are generally required to change coupling frequencies based on C-Coupler1. Moreover, overuse of configuration files can cause problems. Configuration files containing configuration information determined by the model codes can be inconsistent with the model codes. For example, C-Coupler1 will read in the time step of each component model from the runtime configuration files, while the user can change the time step of a component model through the *namelist* file or the model codes. C-Coupler1 will read in each model grid through a grid data file managed by the runtime configuration, while a model grid of a component model can be generated by the model code or read from a grid data file that is not managed by the runtime configuration. To avoid problems resulting from such inconsistencies, extra effort is required to develop code to detect them, and the user will have to fix the corresponding configuration files when an inconsistency is detected.

Therefore, to make the configuration system both user-friendly and flexible, C-Coupler2 should not allow configuration files to include any coupling configuration information determined by component models, and it should provide flexible



APIs to enable component models to specify various coupling configuration information flexibly. Considering that there are various kinds of component models, the configuration system should have commonality in, for example, supporting various kinds of component models and model grids. Considering that the ASCII format can lower the readability of configuration files, another format with better readability should be used to design the configuration files.

## 3.2    Model coupling within one executable or a subset of MPI processes

Similar to CPL6, C-Coupler1 requires each component model to have its own executable. However, there are increased requirements for model coupling within one executable or a subset of MPI processes. For example, CESM (Hurrell et al., 2013) has the component models and driver containing the CPL7 coupler (Craig et al., 2012) enclosed in a unique executable, and any two different component models can run on non-overlapping, partially overlapping, or overlapping MPI processes. The rapid expansion of model codes requires modularization to guarantee the quality of the models' software, and a coupler can be used to achieve this when it can support coupling between different procedures in the same component model.

## 3.3    Dynamic 3-D coupling

Atmospheric chemistry modeling is becoming increasingly important for simulating air quality and climate. Such modeling strongly depends on meteorological fields, and is always included as an internal package in an atmosphere model where the atmospheric chemistry package uses the same 3-D grid as the model. The rapid development of atmospheric chemistry modeling has led to standalone atmospheric chemistry models, such as GEOS-Chem (Long et al., 2015), which read in meteorological fields from data files that can be produced by various atmosphere models. As increasing numbers of atmosphere models require the time-variant aerosol concentration, which can be produced by atmospheric chemistry modeling, there is increasing demand for two-way coupling between an atmosphere model without atmospheric chemistry modeling and a standalone atmospheric chemistry model. Even if an atmosphere model includes an atmospheric chemistry package, considering that atmospheric chemistry modeling generally is very time consuming, it might run with a lower resolution. Overall, there is increasing demand for 3-D coupling between atmosphere models and atmospheric chemistry models (or packages) with different 3-D grids.

Despite its 3-D coupling capability, C-Coupler1 might fail to handle the 3-D coupling between an atmosphere model and an atmospheric chemistry model, because it requires the corresponding 3-D grids to be constant throughout the whole simulation, whereas the terrain-following pressure coordinates that are widely used in atmosphere models and atmospheric chemistry models make the vertical coordinate values of 3-D grids change with the surface pressure in time integration. In this paper, we call 3-D coupling on constant grids "static 3-D coupling" and 3-D coupling on time-variant grids "dynamic 3-D coupling". A coupler having dynamic 3-D coupling capability will be much more capable of achieving coupling between an atmosphere model and an atmospheric chemistry model (or package).




### 3.4  Coupling generator

Model coupling generally involves coupler functions such as data transfer, data interpolation, and data averaging. Most existing couplers require the user to develop explicitly all coupling procedures. This is inflexible and not user-friendly enough, because the user must modify the model code, perhaps even significantly, when developing a new coupled model or

changing coupling configurations. The coupler OASIS (Redler et al., 2010; Valcke, 2013; Craig et al., 2017) is more flexible and user-friendly in this regard, because it can automatically generate coupling procedures.

To make C-Coupler2 flexible and user-friendly, it should also include a coupling generator capable of automatically generating coupling procedures. Although a coupling generator is not included in C-Coupler1, it has already been considered in the overall design of C-Coupler.

### 3.5  Non-blocking data transfer

Data transfer enables a sender to transfer a set of coupling fields to a receiver that is not necessarily in the same component model. A data send/receive operation is blocking when it does not return until the communication is finished (i.e., the receiver has successfully received the data), while a non-blocking operation can return immediately before the communication is finished. In a coupled model, a component model is always both a sender and a receiver (i.e., two-way

coupled), and will execute both data send and receive operations. As mentioned above, C-Coupler aims to enable a model to have identical code versions in different coupled models, so the order of data send and data receive operations in a component model can remain the same in different coupled models. To avoid potential deadlocks, we propose to execute data send operations as early as possible and execute data receive operations as late as possible. Specifically, in the initialization stage or at a time step, data send operations should occur before data receive operations. Fig. 4 shows an

example of model coupling between two component models, in each of which the data send operation is executed before the data receive operation at each time step. During blocking data send/receive operations, the data send operations in both component models cannot return, because the corresponding data receiving operations subsequent to the data send operations will never be executed, leading to a deadlock. Similarly, blocking data transfer can also introduce deadlocks to model coupling within the same component model. Therefore, non-blocking data transfer is highly desirable for developing C-

Coupler2.

### 3.6  Model nesting

Model nesting generally involves nesting a small grid domain with finer resolution into a larger grid domain with coarser resolution. This approach has been widely used in weather forecasting and climate simulation to achieve higher-resolution simulations in key grid domains, without significantly increasing the computational cost. Generally, a regional

model can be nested into another model so that different grid domains are simulated by different models, while some models





such as WRF have self-nesting capability, where different grid domains are simulated by the same model. Although WRF and its self-nesting capability have been widely used, the corresponding software implementation has a number of limitations. First, a data structure that can simultaneously manage the fields on different grid domains and a driver that orders initialization and integration among different grid domains are implemented in WRF. For a regional model without

self-nesting capability, significant code changes in the data structure and driver are required to achieve self-nesting capability. Second, all grid domains must use the same set of MPI processes for integration, so that grid domains must run one by one, not simultaneously. Such an implementation can limit parallelism as well as scalability to the grid domains with fewest grid points, and will also waste parallelism between different grid domains.

Model nesting will introduce field exchange between the same type of component models on different grid domains. As

such field exchange generally includes data transfer and data interpolation that are the fundamental functions of a coupler, model nesting can potentially benefit from couplers. If each domain in model nesting can be treated as a component model in model coupling, a regional model can easily achieve self-nesting with its original data structure only managing the fields on one grid domain, and different domains can be integrated simultaneously on different sets of MPI processes for higher parallelism and better parallel efficiency. To aid in the nesting of a regional coupled model (e.g., a regional ocean–

atmosphere coupled model) to itself or another coupled model, couplers can serve the field exchanges both between the same type of component models on different grid domains and between different types of component models on the same grid domain.

### 3.7    Incremental coupling

Building a new coupled model version involves either directly coupling a set of component models together or updating

an existing coupled model through coupling external component models or replacing some component models. Such updating of an existing coupled model is here called "incremental coupling". Directly coupling many component models to create a new coupled model is difficult and possibly unwise, because it requires much effort in software implementation, software testing, scientific testing, etc., while incremental coupling is always better when a suitable existing coupled model is available. However, incremental coupling may still face some technical challenges when the existing coupled model and

the component models to be coupled have different software frameworks. For example, He et al. (2013) successfully nested WRF into CESM, where both the main driver of CESM and the driver of WRF were modified to enable the main driver of CESM to drive the integration of WRF and so achieve effective nesting. Successful incremental coupling can give a new coupled model that may become a new code version corresponding to the original coupled model. Further developing the original and the new coupled model in separate code version branches can lead to conflicts when trying to merge the two

branches. For example, following the work of He et al. (2013), the main driver of CESM in the original code version branch (managed and maintained by the National Center for Atmospheric Research, NCAR) was significantly changed without



considering WRF nesting, leading to much further work being required to re-nest the latest version of WRF into the latest version of CESM.

As C-Coupler aims to enable a model (either a component model or a coupled model) to have an identical code version in different coupled models (i.e., a model can have the same code in different coupled models after incremental coupling), C-Coupler should be able to improve incremental coupling.

## 3.8 Debugging capability

Models can behave anomalously where they run exits due to an error but without giving a report. In such a case, the corresponding simulation setting might be abandoned and another tried, or much effort might be expended locating and fixing the model code segment corresponding to the abnormal exit. Fixing an error is not easy, because it can easily and quickly propagate throughout a component model and from one component model to another through a coupler.

C-Coupler2 aims to facilitate software debugging for model coupling. Specifically, C-Coupler2 should promptly report an error after an abnormal exit, and the error report should effectively help to locate the code segment or configuration file that requires fixing. Moreover, C-Coupler2 should thoroughly examine its inputs to avoid the propagation of errors.

## 4 Design and implementation of C-Coupler2

As the second version of C-Coupler, C-Coupler2 is guided by the family's general coupling architecture (Fig. 1), so it should be applicable to various coupled models and enable a model to have an identical code version in different coupled models. These considerations influenced the design of the software structure of C-Coupler2 (Fig. 5), which consists of a coupling configuration interface, a coupling generator, and a set of function modules. This software structure is similar to that of the original C-Coupler (Fig. 2), but has the following differences.

1) The original structure of C-Coupler has the coupling generator as a standalone tool that produces the runtime configuration files that drive the runtime software system. However, C-Coupler2 works as a common and flexible library (which can be viewed as the runtime software system), and the coupling generator is an internal program of the library. The coupling generator does not produce runtime configuration files, but directly uses the function modules to generate coupling procedures. Such a design can save redundant code development related to runtime configuration files.

2) Coupling generation in the original structure of C-Coupler fully depends on the offline configuration files that are managed by the configuration system. In C-Coupler2, coupling generation depends on the coupling configuration information obtained by the coupling configuration interface via online API calls and offline configuration files.

3) C-Coupler2 does not include functions to support integrating external algorithms. This will be further discussed in Section 6.



In detail, the function modules of C-Coupler2 include a non-blocking data transfer manager, a component model manager, a grid manager, a remapping manager, a restart manager, a parallel decomposition manager, a time manager, a coupling field instance manager, a coupling interface manager, and a debugging manager. The non-blocking data transfer manager manages a set of runtime data transfer algorithms, each of which is responsible for the non-blocking transfer of a

set of coupling fields within a component model or between two different component models. The component model manager handles basic information (e.g., name, type, MPI processes) about the component models registered to C-Coupler2. The grid manager manages model grids registered to C-Coupler2; similar to the grid manager in C-Coupler1, it also utilizes CoR1 to support various types of grid with dimensions from 1-D to 4-D. The remapping manager controls a set of runtime remapping algorithms, each of which interpolates a set of coupling fields from one grid to another. Similar to the remapping

manager in C-Coupler1, it also utilizes CoR1 to achieve data interpolation between any kind of grid with dimensions from 1-D to 4-D. It has been further upgraded to support dynamic 3-D interpolation. The restart manager helps each component model as well as the whole coupled model to run correct restarts. The parallel decomposition manager oversees parallel decompositions on model grids. Similar to C-Coupler1, each parallel decomposition must be on a 2-D horizontal grid that has been registered to C-Coupler2, while the parallel decomposition on vertical grids remains unsupported. The coupling

field instance manager supervises a set of coupling field instances registered by component models or used by C-Coupler2 internally. The coupling interface manager operates a set of coupling interfaces, each of which imports, exports, or remaps a set of coupling fields. The time manager manages the model time of each component model and manages a set of timers. A timer can be used to control the time to execute a coupling interface and to control lag in model coupling. The debugging manager enables C-Coupler2 as well as component models to flexibly report log information or errors.

We will further introduce here the design and implementation related to each main feature of C-Coupler2, including the common, flexible, and user-friendly coupling configuration interface, model coupling within one executable or a subset of MPI processes, flexible and automatic coupling generation, dynamic 3-D coupling capability, non-blocking data transfer, facilitation for model nesting, facilitation for incremental coupling, and debugging capability.

### 4.1   Common, flexible, and user-friendly coupling configuration interface

The coupling generator of C-Coupler2 can automatically generate coupling procedures for model coupling and nesting. As it takes coupling configuration information as its input, the coupling configuration interface should be able to obtain sufficient information for successful coupling generation. Moreover, the constitution of the coupling configuration information determines the flexibility of specifying coupling configurations, and more types of coupling configuration information generally means greater flexibility of C-Coupler2. Finally, the coupling configuration information of C-

Coupler2 includes the following.

1)   Basic information about each component model, including the model name, model type, the parallel setting (i.e., the MPI processes that are involved in running the component model), and the relationship with other component models.





To facilitate incremental coupling, an existing coupled model with any coupler can be referred to C-Coupler2 as a component model, and a component model of the existing coupled model can be further referred to C-Coupler2 as a child component model. For a component model with self-nesting capability within one executable, one grid domain can be employed as a component model and a smaller grid domain directly nested to it can be used as its child component model. Different component models can share common MPI processes.

2) Coupling connections. Model coupling by C-Coupler2 can be viewed as a set of data flows, each of which couples a set of coupling fields provided by a component model to a component model that uses these fields—possibly the same component model, as C-Coupler2 supports model coupling within one component model. Here we call such a data flow a "coupling connection". The coupling generator can automatically detect all possible coupling connections, while the user can also specify some coupling connections with higher priority.

3) Attributes of coupling fields. Coupling fields are distinguished using field names. All component models in C-Coupler2 share the same name space of the coupling fields as well as the default attributes corresponding to each field name.

4) Model grids. A coupling field is either a scalar variable or locates on a model grid. A model grid may be vertical or horizontal, or a 3-D grid consisting of a horizontal grid and a vertical grid. There might be some relationship between two grids; e.g., a horizontal or vertical grid can be a sub-grid of a 3-D grid.

5) Decomposition of grid domain for parallelization. To accelerate modeling on a modern high-performance computer with many processor cores, a grid domain in a component model is generally decomposed into a number of subdomains, each of which is assigned to an MPI process for parallel integration. We call this "parallel decomposition".

6) Coupling field instances. A coupling field generally has multiple instances in a coupled model. First, different component models can produce or use the same coupling field. For example, when all grid domains in self-nesting WRF are registered as component models in C-Coupler2, they can produce the same coupling fields (e.g., precipitation), where each component model has its own coupling field instances. Second, a given component model can have different instances of the same coupling field due to different model grids or different parallel decompositions on the same model grid. For example, a component model can interpolate a coupling field from a source grid to a target grid, which means that this coupling field has two separate instances: one on the source grid and one on the target grid.

7) Conducting coupling field instances. A component model can export coupling field instances to the coupled model, import coupling field instances from the coupled model, or remap its own coupling field instances on a source grid to the coupling field instances on a target grid.

8) Coupling frequencies. A component model can specify the frequency at which it exports, imports, or remaps the coupling field instances. Different coupling frequencies might be needed in different simulations; e.g., model coupling can be more frequent when the resolution increases.

9) Model time. C-Coupler2 manages model time information for each component model to control model coupling in the time integration of the whole coupled model. It uses a separate and unique time manager for each active component model. The coupling frequencies should be consistent with the model time. For example, a coupling frequency should be




a positive integer multiple of the time step of the corresponding model.

10) Remapping configurations. Most existing couplers, including C-Coupler1, enables the user to specify how to remap a set of fields from a source grid to a target grid; e.g., using the offline remapping weights read from an input data file produced by a remapping software tool or using the online remapping weights produced by the coupler (if supported).

11) Shared input parameters for a model run. C-Coupler will require shared input parameters for a model run, such as case information of the model run, the start time of the model run, how to stop the model run, and the frequency at which to write restart files.

The above coupling configuration information can be classified into two categories: private coupling configuration information of a component model (including information about the component model, model grids, parallel decompositions,

coupling field instances, conducting coupling field instances, coupling frequencies, and model time) and public coupling configuration information shared by component models (including coupling connections, attributes of coupling fields, and shared input parameters for a model run). Considering the motivation for coupling configuration (Section 3.1), we design a set of C-Coupler2 APIs to enable a component model to specify flexibly its private coupling configuration information through model codes, and design a set of configuration files for flexibly specifying the public coupling configuration

information. Although the remapping configurations can be either private or public (the source and target grid used in data remapping may belong to the same or different component models), we only design the corresponding configuration file to guarantee a unique way to specify remapping configurations.

We introduce in this section the implementation of the coupling configuration interface for each kind of configuration information.

**4.1.1   C-Coupler2 APIs**

**4.1.1.1   APIs for component model management**

To couple component models running on non-overlapping, partially overlapping, or overlapping subsets of MPI processes, C-Coupler2 allows a component model to run on any subset of MPI processes. Therefore, the coupler can support almost any kind of MPI process layout among the component models. Figure 6 shows an example of a complex MPI process

layout: *comp1*, *comp2*, and *comp3* do not share any MPI process; *comp4* runs on a proper subset of the MPI processes of *comp1*; *comp8* run on all MPI processes of *comp2*; and *comp4* and *comp5* partially share some MPI processes. Moreover, there are relationships between the component models in Fig. 6: *comp1* is the parent of *comp4* and *comp5*; *comp5* is the parent of *comp6* and *comp7*; and *comp2* is the parent of *comp8*. In C-Coupler2, a component model must cover all MPI processes of its children (e.g., *comp1* in Fig. 6 includes all processes of *comp4* and *comp5*). A component model without a

parent is a root component model (e.g., *comp1*, *comp2*, and *comp3* in Fig. 6 are root component models). Each MPI process



must belong to a unique root component model (e.g., each process in Fig. 6 only belongs to one of *comp1*, *comp2*, or *comp3*); i.e., all root component models cover all MPI processes without sharing any MPI process with each other. This constraint seems contradictory to the target of supporting shared MPI processes among component models, and may make C-Coupler2 unable to support some MPI process layouts. For example, given that a component model consists of two component models

that run on partially overlapping subsets of MPI processes, both component models cannot be root component models. To support this kind of MPI process layout, a coupled model can be registered as a root component model of C-Coupler2, and its component models can be further registered as children of the root component model.

The APIs for component model management are listed in Table 1. The API "*CCPL_register_component*" is responsible for registering a component model to C-Coupler2. C-Coupler2 only serves component models registered to it. (A component

model whose model coupling is fully served by other couplers but not C-Coupler2 is unnecessarily registered to C-Coupler2). The arguments of this API include the ID of the parent component model, model name, model type, and MPI communicator. Any component model except a root component model must have a parent. C-Coupler2 will allocate an ID and generate a unique full name for each component model that is formatted as "*parent_full_name@model_name*", where "*model_name*" means the name of the current component model and "*parent_full_name*" is the full name of the parent component model.

(For a root component model, "*parent_full_name*" corresponds to an empty string.) A component model is either active or pseudo (inactive), as specified by the model type. A pseudo component model can be the parent of some component models, while its name will not be included in the full name of any component model. Moreover, coupling configurations cannot be further specified to a pseudo component model. Table 2 lists the model types currently supported by C-Coupler2. Note that "active_coupled_system" and "pseudo_coupled_system" indicate that an existing coupled model can be registered as a

component model of C-Coupler2. This API can create the MPI communicator of the component model when required. It will start the stage of coupling configuration of the component model, while the API "*CCPL_end_coupling_configuration*" will finalize the stage of coupling configuration. A component model can successfully call "*CCPL_end_coupling_configuration*" only when all its children component models have already called this API.

For more details about the APIs listed in Table 1, please refer to the user guide (https://gitlab.com/c-coupler-group/c-

25 coupler-doc/raw/master/C-Coupler2%20User%20Guide.pdf).

### 4.1.1.2 APIs for time management

Table 3 lists the APIs for time management that enable C-Coupler2 to manage the model time information for each active component model. Detailed time information of a component model can also be accessed through C-Coupler2, and thus a component model can employ C-Coupler2 for its model time management. A component model with its own model

time management must keep its model time constantly consistent with C-Coupler2. The API "*CCPL_check_current_time*" can be used to check such consistency. An active component model can have a unique time manager that is not activated until a unique time step has been set through the API "*CCPL_set_time_step*". After a time manager is activated, the user can





access detailed information on the model time, define timers, advance the model time, and use timers to control model coupling.

C-Coupler2 currently only provides the API "*CCPL_define_single_timer*" to define a periodic timer. The arguments of this API include the ID of the corresponding component model, a period unit, a period count, a local lag count, and an

optional remote lag count. The period unit and period count specify the period of the timer. The local lag count corresponds to the period unit, which is used to specify a local lag (it can be viewed as a time offset from the start time) that influences when the timer is on. For example, a timer set with <*period_unit*="steps", *period_count*="5", *local_lag_count*="2"> will be on at the $2^{nd}$, $7^{th}$, $12^{th}$, etc. (i.e., $5i + 2$, where $i$ is a non-negative integer) time steps of the corresponding component model. The remote lag count also corresponds to the period unit. It can be used to specify a lag on a coupling connection between

two component models or within one component model. Its default value is 0 (i.e., no lag). Note that the lag for a coupling connection is determined by the timer from the receiver component model. The lag corresponding to a coupling connection can be viewed as the model time difference from the receiver component model to the sender component model, which can control the time sequence between the two component models. For example, given a lag of 1/-1 hour, the coupling fields produced by the sender component model at the sender's $0^{th}/1^{st}$ hour will be obtained by the receiver component model at the

receiver's $1^{st}/0^{th}$ hour. Thus, the user can flexibly achieve concurrent run or sequential run between component models. Incorrectly setting "*remote_lag_count*" may introduce deadlocks between component models.

For more details about the APIs listed in Table 3, please refer to the user guide.

### 4.1.1.3  APIs for grid management

Each grid managed by C-Coupler2 belongs to a unique active component model. A grid shared by multiple component

models should be registered to each component model separately. The keyword for a grid can be expressed as <*ID of the component model*, *grid name*>. Therefore, different grids in the same component model cannot have the same grid name, while grids in different component models can have the same grid name.

Table 4 lists the APIs for grid management. A horizontal grid can be registered via global grid data (through the API "*CCPL_register_H2D_grid_via_global_data*"),            local          grid          data          (through          the          API

"*CCPL_register_H2D_grid_via_local_data*"), or a grid data file (through the API "*CCPL_register_H2D_grid_via_file*"). Considering that a horizontal grid in a component model may be determined by another component model (e.g., the horizontal grid of a land surface model will be determined by an atmosphere model when both models require the same horizontal grid), we designed the API "*CCPL_register_H2D_grid_from_another_component*". A vertical grid can be registered via global grid data. The coordinate of a vertical grid can be registered as a Z coordinate (through the API

"*CCPL_register_V1D_Z_grid_via_model_data*"),          a          SIGMA          coordinate          (through          the          API "*CCPL_register_V1D_SIGMA_grid_via_model_data*"),          or          a          HYBRID          coordinate          (through          the          API "*CCPL_register_V1D_HYBRID_grid_via_model_data*"). A 3-D grid can be registered by combining a horizontal grid and a




vertical grid (through the API *CCPL_register_MD_grid_via_multi_grids*). Thus, C-Coupler2 can know the relationship between a 3-D grid and its sub grids. A 3-D grid can be either an interface-level grid or a middle-point grid. A middle-point grid can be generated from an interface-level grid through the API "*CCPL_register_mid_point_grid*". Thus, C-Coupler2 can know the relationship between an interface-level 3-D grid and a middle-point 3-D grid.

For a 3-D grid that consists of a horizontal grid and a vertical grid with SIGMA or HYBRID coordinate, C-Coupler2 can set its unique surface field on the horizontal grid in order to calculate the vertical coordinates at each horizontal grid point. The surface field of a 3-D grid can be static (through the API "*CCPL_set_3D_grid_constant_surface_field*"), dynamic (through the API "*CCPL_set_3D_grid_variable_surface_field*"), or external (through the API "*CCPL_set_3D_grid_external_surface_field*"). A static surface field has constant values with time integration, so the

vertical coordinate values in the corresponding 3-D grid are constant. A dynamic surface field has changing values with time integration, so that the vertical coordinate values in the corresponding 3-D grid vary. An external surface field has values determined by the surface field of another 3-D grid.

    For more details about the APIs listed in Table 4, please refer to the user guide.

#### 4.1.1.4  API for parallel decomposition management

To accelerate modeling by taking advantage of a high-performance computer with many processor cores, the model needs to be parallelized with MPI, whereby the domains of the model grids are decomposed into separate subdomains for parallel integration. To accommodate the parallel integration of component models and to allow model coupling to be handled in parallel, C-Coupler2 both manages parallel decompositions and provides APIs to enable active component models to register their parallel decompositions to the coupler. C-Coupler2 currently supports parallel decompositions only

on horizontal grids, and further parallel decomposition on vertical grids is yet not supported. Therefore, parallel decomposition is associated with a horizontal grid, and thus with the component model corresponding to the horizontal grid. The keyword for parallel decomposition is expressed as <*ID of component model*, *parallel decomposition name*>. Therefore, different parallel decompositions in the same component model cannot have the same name, while parallel decompositions in different component models can have the same name. Multiple parallel decompositions on the same horizontal grid are

allowed.

    Parallel decomposition on a horizontal grid is described through enumerating global grid cell indexes of the local grid cells assigned to each MPI process of the corresponding component model. A valid global grid cell index should be between 1 and the size of the horizontal grid. For any local grid cells that need not be considered in model coupling (e.g., land-only grid cells in an ocean model), the corresponding values of the global grid cell index can be set to a C-Coupler2 pre-defined

variable *CCPL_NULL_INT*, to save some overheads in model coupling.

    Table 5 lists the unique API for parallel decomposition management. Please refer to the user guide for more details.



### 4.1.1.5 API for coupling field instance management

A coupling field instance includes a set of meta information and a memory buffer that keeps the data values of an instance of a coupling field. A coupling field instance is associated with a unique component model, a unique grid, and a unique parallel decomposition. An attribute of "*buf_mark*" is employed in each coupling field instance to separate multiple

coupling field instances in the same component model, on the same grid, and on the same parallel decomposition. For example, as the land surface, oceans, and sea ice lie under the atmosphere, an atmosphere model may receive multiple coupling field instances of surface temperature from land surface, ocean, or sea ice models. Therefore, the keyword for a coupling field instance is expressed as <*field name*, *ID of component model*, *ID of grid*, *ID of parallel decomposition*, *buf_mark*>. For a scalar coupling field instance that is not on a grid, the corresponding grid ID and parallel decomposition ID

should be set to *-1*. The C-Coupler2 APIs allow a component model to register field instances to the coupler to provide, obtain, and remap coupling field instances in model coupling.

Table 6 lists the unique API for coupling field instance management. Please refer to the user guide for more details.

### 4.1.1.6 APIs for coupling interface management

In C-Coupler2, an active component model can handle coupling field instances through coupling interfaces. The

15 keyword of a coupling interface is expressed as <*ID of the component model*, *interface name*>. Therefore, different coupling interfaces in the same component model cannot have the same interface name, while coupling interfaces in different component models can have the same interface name.

Coupling interfaces are classified into three categories: import, export, and remap. An import interface enables a component model to obtain coupling field instances from itself or other component models. Specifically, it can be specified

to obtain instantaneous or averaged coupling field instances. An export interface enables a component model to provide a number of coupling field instances to the coupled model. A remap interface enables a component model to remap its coupling fields from a source grid to a target grid. There are two detailed kinds of remap interface: normal and fraction based. A normal remap interface directly interpolates coupling field instances from the source grid to the target grid, while a fraction-based remap interface will first adjust the values of coupling field instances on the source grid based on the source

fraction before remapping and finally adjusting the values of coupling field instances on the target grid based on the target fraction after remapping. (The source fraction is also remapped from the source grid to the target grid to produce the target fraction at the same time.) Fraction-based remap interfaces are generally necessary to guarantee conservation in model coupling.

There are three steps taken to utilize a coupling interface. The coupling interface is first registered, whereby a timer is

30 required to be specified to control the timing of coupling interface execution. Coupling procedures are next generated for the coupling interface, which is then executed in the third step. Although the API to execute a coupling interface can be called at each time step, a coupling interface will be truly executed only when its timer is bypassed or its timer is on. C-Coupler2





allows the timer to be bypassed when executing a coupling interface, in order to achieve flexible coupling at the initialization stage of the coupled model. Note that the timer of a coupling interface cannot be bypassed again if this coupling interface has already been executed with the timer on, and when the timer of a coupling interface is not bypassed, the coupling interface will be truly executed at most once each time step, which means that any additional API calls for executing the coupling

interface at a time step will be ignored.

For a remap interface that does not refer to coupling between different coupling interfaces or different component models, its coupling procedures are generated implicitly by the coupling generator when registering it. Coupling procedures of an export/import interface are also generated automatically by the coupling generator, but will not be generated when registering the interface, because an export/import interface refers to coupling between different coupling interfaces in the

same or different component models. To generate coupling procedures for export or import interfaces, the coupling generator will analyze possible connections from export interfaces to import interfaces based on the field name of each coupling field instance. A coupling connection from an export interface to an import interface can be generated only when these two coupling interfaces have common field names. Regarding a field name, C-Coupler2 allows an export interface to be connected to any number of import interfaces, while forcing an import interface to be connected from a unique export

interface. In other words, each coupling field instance in an import interface must have only one provider. If there are multiple providers for a coupling field instance in an import interface, the user must select only one provider through the corresponding configuration file (see Section 4.1.2.4). Different coupling field instances in an import interface can have different providers. The coupling procedures for import and export interfaces are generated through explicitly calling the APIs for coupling generation.

An export interface or a remap interface can always be executed successfully without error, while the execution of import interfaces can fail and lead to an error report, if the coupling procedures of some necessary coupling field instances have not been generated (i.e., if the providers of some necessary coupling field instances have not been found). When registering an import interface (through the API "*CCPL_register_import_interface*"), each import coupling field instance can be specified as necessary or optional. No error will be reported if the providers of some optional coupling field instances

have not been found.

Table 7 lists the APIs for coupling interface management. Please refer to the user guide for more details.

### 4.1.1.7  APIs for coupling generation

Two designs for the coupling generation function were compared. The first enforces only one coupling generation for the whole coupled model: the unique coupling generation is performed when finalizing the coupling configuration stages of

the whole coupled model (when all root component models are calling the API "*CCPL_end_coupling_configuration*"), and it involves all component models that have already been registered to C-Coupler2. This unique global coupling generation function was not favored for two main reasons, as follows.





1) This design assumes that the whole coupled model as well as each component model is organized as a unique three-stage flowchart consisting of a coupling configuration stage, coupling generation, and a model coupling run. However, many existing models such as CESM consist of not a unique but multiple three-stage flowcharts, indicating that multiple coupling generations are necessary for wide usage in real cases.

2) As a global coupling generation will involve the participation of all component models, and will require global synchronization of all the MPI processes in the whole coupled model, it will be costly, inconvenient, and unnecessary to conduct multiple global coupling generations. For example, an increment coupling case that seeks to nest a regional atmosphere model into an existing climate system model through C-Coupler2 requires only partial coupling generation between the regional atmosphere model and the global atmosphere model.

The second design achieves partial coupling generation for any subset of component models at any time through the APIs for coupling generation listed in Table 8. The coupling generation related to a component model is classified as either individual or family. Individual mode considers only the given component model in coupling generation, while family mode considers the given component model and its descendant component models in the same coupling generation. When registering a component model through the API "*CCPL_register_component*", it can be specified to enable or disable the given component model in the family coupling generation of its parent or any ancestor. The API "*CCPL_do_external_coupling_generation*" can do coupling generation regarding any subset of component models, where either individual or family coupling generation can be specified for each given component model. The API "*CCPL_get_configurable_comps_full_names*" allows flexible specification of a subset of component models in an XML configuration file; it can cooperate with the API "*CCPL_do_external_coupling_generation*" to improve further the flexibility of coupling generation. Besides partial coupling generations, a global coupling generation will still be performed when root component models are calling the API "*CCPL_end_coupling_configuration*", while a root component model that has been disabled in the family coupling generation will not be involved in the global coupling generation.

Coupling generation requires the synchronization of all MPI processes of the involved component models. Please refer to the user guide for more details of the APIs for coupling generation.

### 4.1.1.8 Other APIs

Like most component models, C-Coupler2 can restart model simulation from a checkpoint. It does so through three APIs: *CCPL_do_restart_write_IO*, *CCPL_do_restart_read_IO*, and *CCPL_is_restart_timer_on* (Table 9). Currently, the restart management only serves local variables or data managed by C-Coupler2. Therefore, the restarting of component model variables (including component model variables registered to C-Coupler2 as coupling field instances) must be managed by the component models themselves. To restart a model coupled using C-Coupler2, all active component models



should separately call the two APIs *CCPL_do_restart_write_IO* and *CCPL_do_restart_read_IO*. Besides the "*initial*" run, C-Coupler2 supports three types of model run: "*continue*", "*branch*", and "*hybrid*", which are related to the restart capability.

C-Coupler2 enables each MPI process in each component model to have a separate log file, thus improving parallel debugging capability. Several APIs ("*CCPL_report_log*", "*CCPL_report_progress*", "*CCPL_report_error*",

"*CCPL_get_comp_log_file_name*" and "*CCPL_get_comp_log_file_device*") allow component models to benefit from such a capability.

For more details of the APIs in Table 9, please refer to the user guide.

### 4.1.1.9  Examples of implementing a coupled model with C-Coupler2 APIs

Figure 7 shows an example of the use of C-Coupler2 APIs to achieve hybrid coupling configuration and model coupling

during the initialization stage of a coupled model with four component models (*comp1–comp4*). We assume that *comp1* and *comp2* are coupled together, *comp3* and *comp4* are coupled together, and that *comp3* and *comp4* are the children of *comp1* and depend on some boundary conditions from *comp1*. First, *comp1* and *comp2*, which cover all MPI processes (processes 0–34) and do not share any MPI process, simultaneously call the API *CCP_register_component* to register themselves as the root component models. The child component models *comp3* and *comp4* partially share a subset of MPI processes (processes

9–12). All MPI processes of *comp3* first register *comp3* as a child of *comp1*, and next set the unique time step, register several model grids, register a parallel decomposition, register several coupling field instances, specify a coupling field instance as the dynamic surface field of a 3-D grid, define several timers, and register several coupling interfaces. After calling the API "*CCPL_do_individual_coupling_generation*" for coupling generation within *comp3* itself, *comp3* executes some coupling interfaces, and then finalizes its coupling configuration stage through calling the API

"*CCPL_end_coupling_configuration*". *Comp4* follows a C-Coupler2 flowchart similar to *comp3*. As *comp3* and *comp4* share some processes, they cannot conduct coupling configuration and model coupling at the same time in most cases (in this example, we specify *comp3* to run before *comp4*), except for the simultaneous calling of the API "*CCPL_do_external_coupling_generation*" that can generate coupling procedures for the coupling connections between the two child models. After both child models have finished their coupling initialization stage, their parent conducts its coupling

configuration, following a similar flowchart. As *comp1* shares processes with its children, *comp1* cannot conduct coupling registration simultaneously with *comp3* and *comp4*, and thus *comp1* runs after its children here. As *comp2* does not share any process with the other component models, it can conduct coupling registration simultaneously with *comp1*, *comp3*, and *comp4*. Finally, *comp1* and *comp2*, the root component models, simultaneously call the API "*CCPL_end_coupling_configuration*" to finalize the coupling configuration stage of themselves and the whole coupled

model and to invoke global coupling generation. At the end of the initialization stage, each component model can read in the restart data files when necessary.



Figure 8 shows an example of model coupling in the kernel (time integration) stage of the coupled model in Figure 7. In addition to the assumptions in Figure 7, *comp1* and *comp2* are further assumed to have the same time step, which is double that of *comp3* and *comp4*. All coupling interfaces are executed here without bypassing the timers. At a time step of *comp1* and *comp2*, they can simultaneously execute coupling interfaces, call the API "*CCPL_do_restart_write_IO*" to generate
restart data files when the restart timer is bypassed or is on, and finally call the API "*CCPL_advance_time*" to advance the model time managed by C-Coupler2. We strongly recommend checking the consistency of model time between a component model and C-Coupler through calling the API "*CCPL_check_current_time*". *Comp3* and *comp4* alternately use a C-Coupler2 flowchart similar to that for *comp1* and *comp2*, but they will advance their model time twice when *comp1* and *comp2* advance their model time once.

### 4.1.2 C-Coupler2 configuration files

As mentioned above, the C-Coupler2 configuration files allow flexible specification of public coupling configuration information including shared input parameters for a model run, attributes of coupling fields, remapping configurations, and coupling connections. In order to achieve good readability, all configuration files are in XML format. This subsection briefly introduces the four kinds of configuration files; i.e., input parameter configuration file, field attribute configuration file,
remapping configuration file, and coupling connection configuration file. Additional details can be found in the user guide.

#### 4.1.2.1 Input parameter configuration file

The input parameter configuration file specifies a set of global input parameters shared by all component models. The input parameters include simulation times (e.g., start and stop times), the type of simulation run (i.e., initial, continue, branch, or hybrid), whether leap years are considered, and the frequency of writing restart data files. Note that C-Coupler2 requires
all component models to use the same start and stop times, and the user should guarantee that the input parameter configuration file is consistent with the corresponding modeling settings.

#### 4.1.2.2 Field attribute configuration file

When registering a coupling field instance to C-Coupler2, the field name should be specified as an input parameter. A field name is legal only when there is a corresponding entry in the field attribute configuration file that is shared by all
component models in a coupled model. When the coupling generator tries automatically to generate coupling procedures, field names are used to detect possible coupling connections between coupling interfaces: an import interface and an export interface can have a coupling connection only when their coupling field instances have common field names. The attributes of each coupling field include "*long_name*", "*default_unit*", "*dimensions*", and "*type*". The attribute "*dimensions*" means a label of grid dimensions. It can be set to "*0D*", "*H2D*", "*V1D*", or "*V3D*", denoting that a field is a scalar variable that is not



on any grid, is on a horizontal grid, is on a vertical grid, or is on a 3-D grid that consists of a horizontal grid and a vertical grid, respectively. The attribute "*type*" gives the type of the coupling field, either "*state*" or "*flux*".

### 4.1.2.3 Remapping configuration file

Remapping configuration files can flexibly and conveniently specify the remapping of coupling fields between grids, as
follows.

1) For remapping from a source horizontal grid to a target horizontal grid, the user can either employ the remapping weights that are automatically generated by C-Coupler2 in parallel or read from an existing remapping weight file produced by external software such as SCRIP, ESMF, YAC, or CoR.

2) Like C-Coupler1, C-Coupler2 uses the "2-D + 1-D" approach to achieve 3-D remapping. Regarding 3-D remapping, the
remapping configuration for the 2-D (horizontal) remapping and for the 1-D (vertical) remapping can be specified separately; the 2-D remapping can also use the remapping weights loaded from a remapping weight file.

3) Different coupling fields in the same component model can have different remapping configurations, and the same coupling field in different component models can have different remapping configurations.

4) Given a coupling field, a component model can either use its own remapping configuration or use that inherited from its
parent (if its own remapping configuration is not specified); a root component model (i.e., one without a parent) without a specified remapping configuration can use the specified overall remapping configuration or use the default remapping configuration set by C-Coupler2 (if the overall remapping configuration is not specified). In the default remapping configuration, the bilinear remapping algorithm is used to remap the "*state*" fields between horizontal grids, the conservative remapping algorithm is used to remap the "*flux*" fields between horizontal grids, and the linear remapping
algorithm is used to remap the vertical and time dimensions. Note that all remapping weights in the default remapping configuration are generated automatically by C-Coupler2.

5) A remapping configuration file consists of a set of remapping settings, each of which can specify the remapping configuration for all coupling fields, coupling fields of the same type ("*flux*" or "*state*"), or a specific set of coupling fields (possibly even only one field). A prioritization strategy is designed accordingly: a remapping setting
corresponding to all coupling fields is at the lowest priority, a remapping setting corresponding to a type of coupling fields is at medium priority, and a remapping setting corresponding to specific coupling fields is at the highest priority.

A procedure with data remapping for a given coupling field on a coupling connection between two different component models will be generated when the component models use different corresponding grids. It is possible that the remapping configuration of this coupling field is not the same in the two component models. In such a case, C-Coupler2 will only use
the remapping configuration in the source component model (the component model that exports the coupling field). In general, given a coupling field on a coupling connection, C-Coupler2 uses only the remapping configuration in the source



component model for coupling generation. Therefore, it is meaningless to specify remapping configurations for the imported coupling fields of a component model.

Figure 9 shows an example of a remapping configuration file that consists of three active remapping settings (corresponding to the XML node of "*remapping_setting*" with the attribute "*status*" of value "*on*"). The first remapping

setting (from L1 to L15 in Fig. 9) corresponds to all coupling fields, the second remapping setting (from L16 to L27 in Fig. 9) corresponds to the "*flux*" coupling fields, and the third (from L28 to L39 in Fig. 9) corresponds to two specific coupling fields "*t_atm_3D*" and "*ghs_atm_3D*" that should be 3-D "*state*" fields. Specific to the remapping configuration of these two fields, the vertical remapping configuration is determined by the third remapping setting and the horizontal remapping configuration is determined by the first remapping setting. Both the first and second remapping settings specify an online

horizontal remapping algorithm (corresponding to the XML node of "*H2D_algorithm*") and offline horizontal remapping weight files (corresponding to the XML node of "*H2D_weights*"). Note that offline remapping weight files have higher priority than the online remapping algorithm. To generate the coupling procedures for data remapping, a remapping weight file in the corresponding remapping configuration (if present) will be used if it matches both the source grid and the target grid of this data remapping.

**4.1.2.4  Coupling connection configuration files**

A coupling connection configuration file can be used to specify coupling connections for a component model. The connections are classified into three types: 1) for import interfaces, 2) for model grids (corresponding to the API "*CCPL_register_H2D_grid_from_another_component*"), and 3) sets of component model full names used for external coupling generation (corresponding to the API "*CCPL_do_external_coupling_generation*"). The coupling connections for an

import interface specify the providers (a provider is a component model as well as its export interface) of imported coupling fields, while a distinct subset of imported coupling fields can have a distinct provider. When the coupling generator tries to generate coupling procedures for an imported field of an import interface, it will first check the corresponding coupling connection configuration file. If the configuration file contains a coupling connection for the imported field, the coupling generator will only use the corresponding provider in coupling generation; otherwise, it will try to detect a provider and will

report an error when no provider or multiple providers are detected.

**4.2      Model coupling within one executable or a subset of MPI processes**

As mentioned in Section 4.1, to support model coupling within one executable or a subset of MPI processes, C-Coupler2 allows a component model to run on any subset of MPI processes and can generate coupling connections between the export and import interfaces of the same component model. Each component model registered to C-Coupler2 can have

its own model coupling resources, including time step, timers, model grids, parallel decompositions, coupling field instances, and coupling interfaces. In other words, a model coupling resource must be associated with a unique component model.



Most model coupling resources, including the time step, model grids, parallel decompositions, coupling field instances, and coupling interfaces, are public to a component model and shared by all its MPI processes. When registering a public model coupling resource of a component model, all MPI processes of the component model are required to call the corresponding API simultaneously, with consistent parameters. To manage different component models and model coupling resources

effectively, each component model, as well as each model coupling resource, has a unique ID.

Note that OASIS3-MCT_3.0 can also achieve model coupling within one executable or a subset of MPI processes, while its implementation is different from C-Coupler2. In OASIS3-MCT_3.0, different component models can share the same executable but cannot share an MPI process, while model grids and parallel decompositions can exist across non-overlapping, partially overlapping, or overlapping MPI processes. CESM allows any two different component models to run

on non-overlapping, overlapping, or partially overlapping MPI processes, and should be treated as a unique component model of OASIS3-MCT_3.0; any coupling between different component models of CESM should be treated as a coupling between different grids of the single component model of OASIS3-MCT_3.0. In contrast, each component model of CESM can be treated as a component model of C-Coupler2, and coupling between different component models of CESM can still be treated as coupling between different component models. Direct comparisons of these implementation are difficult, but

the implementation in C-Coupler2 provides essential help to manage model coupling resources effectively, achieve flexible model coupling, improve parallel debugging, and facilitate model nesting and increment coupling.

### 4.3    Flexible and automatic coupling generation

As mentioned in Section 4.1.1.7, C-Coupler2 can achieve partial coupling generation for any subset of component models at any time. Given a subset of component models that participate in a partial coupling generation, the first challenge

is to make a component model know the MPI processes of each other component model, because the partial coupling generation will introduce MPI communications within the MPI processes of the subset of component models, and should not introduce communication with the remaining MPI processes of the whole coupled model. As the registration or initialization of root component models generally involves all MPI processes of the whole coupled model, each root component model can easily know the MPI processes of any other root component model through MPI communications such as global

synchronizations. This may not present a challenge to C-Coupler1 or OASIS3-MCT_3.0, because there are only root component models in the whole coupled model. However, it causes a problem for C-Coupler2, because there are non-root component models (e.g., *comp4–comp8* in Fig. 6). As the registration of a non-root component model only involves a subset of MPI processes (the MPI processes of itself in general and the MPI processes of its parent component model at most), it is almost impossible to guarantee that a component model knows the information of all other existing component models

through MPI communications. To overcome this challenge, C-Coupler2 writes information about the MPI processes of a component model into an internal XML file when registering the component model. Thus, a component model can know the



MPI processes of any other existing component model through the corresponding XML file. Only one MPI process of a component model will write or read the XML file, to minimize the overhead of this implementation.

The coupling generator will generate a coupling procedure for each coupling connection that couples a subset of fields from an export interface to an import interface. As introduced in Section 4.1.1.6, the API for registering an import/export

interface takes as an input parameter a timer, which specifies when a component model must import/export coupling fields. The second challenge during coupling generation is achieving effective coupling when the timers of the import and export interfaces, which can be set independently, are different or even do not "match" in periods. For example, how to achieve effective coupling when the periods of the import and export timers are 900 and 200 seconds respectively (assumed that no lags are specified in the two timers)? At the model time of 0 s, both timers are activated, and the import interface will obtain

the coupling fields from the export interface. Before the second activation of the import timer (at 900 s), the export timer will have been on four times (i.e., at 200, 400, 600, and 800 s). The import interface at 900 s can obtain either the average values of the coupling fields from the four times at which the export timer was on, or the instantaneous coupling fields at its last activation (i.e., at 800 s). Similarly, before the third activation of the import timer (at 1800 s), the export timer will have been on a further five times, at model times of 1000, 1200, 1400, 1600, and 1800 s. Thus, the import interface (at 1800 s) will

obtain from the export interface either the average values of the coupling fields at its five intervening occasions or the instantaneous coupling fields at its last activation (at 1800 s). As an additional example, suppose swapping the periods of the import and export timers (i.e., 200 and 900 s, respectively), still with no lags specified in the timers. As before, the import interface will obtain the coupling fields from the export interface at the model time of 0 s when both timers are on. Although the import timer will be further activated at model times of 200, 400, 600 and 800 s, the import interface will not obtain new

coupling fields from the export interface, but will use those previously obtained at 0 s, because the export interface will not export coupling fields again until a model time of 900 s. In summary, regardless of the difference in periods between the import and export timers, C-Coupler2 can adapt to conduct model coupling in a suitable manner.

As introduced in Sections 4.1.1.2 and 4.1.1.6, C-Coupler2 enables to specify a lag in model coupling via the timer of an import interface. Given a lag of $m$ seconds ($m \neq 0$), the coupling fields obtained by an import component model at its model

time of $N + m$ seconds are exported by an export component model at its model time of $N$ seconds. To support lags, existing coupled models such as CCSM3 (Collins et al., 2006) or FGOALS-g2 essentially extend the simulation stop time of some component models, so that component models have different simulation periods. However, C-Coupler2 only supports a uniform simulation period among all component models. To support lags in C-Coupler2 effectively, an import interface is not executed if it would request coupling fields produced after the simulation stop time, and an export interface is not

executed if its coupling fields would be used after the simulation stop time. Such an implementation may introduce errors into the model states at the last steps of simulation. We therefore propose to extend the simulation period to guarantee correct simulation of the model states in the concerned simulation period.

The coupling generation for a subset of component models follows the steps outlined below.

1)   No matter which API is used to start the coupling generation (Section 4.1.1.7), the coupling generator first confirms the





subset of component models participating in the coupling generation and confirms their MPI processes.

2) Determine all coupling connections. An export interface and an import interface will be connected for model coupling only when they have common coupling fields (with the same field names). As a component model manages its own coupling interfaces as well as coupling fields, an MPI communicator that includes all MPI processes in the subset of component models will be generated for aggregating the information of all coupling interfaces among different component models. As the user can also specify coupling connections through configuration files, file reading is required for analyzing possible coupling connections. To minimize the cost of reading, only one MPI process analyzes possible coupling connections, while other MPI processes await its results. An error will be reported if a coupling field in an import interface has multiple providers in this coupling generation.

3) Generate a coupling procedure for each coupling connection. A coupling connection aims to couple a set of fields from an export interface to an import interface. When these interfaces belong to different component models, their models will exchange information about the corresponding timers, model grids, parallel decompositions, remapping configurations, data types, etc. If a coupling field has different data types in the two coupling interfaces, an operation of data type transformation will be generated. The coupling generator adaptively selects a component model to execute the data type transformation for improved model coupling. For example, given that the data type in the export/import interface is *double*/*float*, the export component model will transform the data type from *double* (8 bytes) to *float* (4 bytes), so that *float* values but not *double* values will be transferred from the export interface to the import interface. If a coupling field has different grids in the two coupling interfaces, a runtime algorithm for parallel data interpolation will be generated following the corresponding remapping configuration, where existing remapping weights will be used or new remapping weights will be read from an external data file or calculated by C-Coupler2 online and in parallel. Currently, only the import component model executes the parallel data interpolation. In the future, the coupling generator will adaptively select a component model to process data interpolation calculation for better coupling performance. When the import interface has been specified to import time-averaged coupling fields, operations for data averaging will be generated. To transfer the coupling fields from the export interface to the import interface, a runtime algorithm of non-block data transfer will be generated. In summary, a coupling procedure can include as necessary a runtime algorithm for data transfer, a runtime algorithm for data interpolation, operations for data type transformation, and operations for data averaging.

As a coupling generation can be performed at any time for any subset of component models, a component model can participate in multiple coupling generations. In other words, the coupling procedures of a component model or even an import/export interface can be incrementally generated through multiple coupling generations. For an import interface in a coupling generation, only the import fields whose coupling procedures have not been generated will be considered in the coupling generation, while the import fields whose coupling procedures have already been generated will be neglected.





## 4.4 Dynamic 3-D coupling capability

Given a 3-D grid that consists of a horizontal grid and a vertical grid with SIGMA or HYBRID coordinates, the vertical coordinate values at each horizontal grid point are determined by a unique surface field on the horizontal grid. For example, the 3-D grid of an atmosphere model with SIGMA or HYBRID coordinates will have constant vertical coordinate values
when the surface field is terrain height, but the values will be variable and change in time integration when the surface field is surface pressure, because the terrain height generally remains constant while the surface pressure changes in time integration. C-Coupler2 therefore provides two APIs, "*CCPL_set_3D_grid_constant_surface_field*" and "*CCPL_set_3D_grid_variable_surface_field*" (Section 4.1.1.3), for specifying constant and variable surface fields, respectively. Given a 3-D grid of an atmospheric chemistry model, the API "*CCPL_set_3D_grid_variable_surface_field*"
can be used to specify the surface pressure as the surface field. As an atmospheric chemistry model generally does not produce the surface pressure, additional implementation will be required to enable an atmospheric chemistry model to obtain external surface pressure (e.g., from an atmosphere model in online model coupling). To facilitate coupling implementation for such a case, C-Coupler2 provides the API "*CCPL_set_3D_grid_external_surface_field*", which, rather than specifying a surface field, states that the surface field of a 3-D grid is externally determined by the surface field of another 3-D grid.
Moreover, the external surface field of a 3-D grid will be obtained automatically and implicitly by C-Coupler2 in model coupling.

As mentioned above, the 3-D interpolation involved in 3-D coupling is still performed in the "2-D + 1-D" manner in C-Coupler2, where 2-D interpolation between the horizontal sub grids is performed first, followed by 1-D coupling between the vertical sub grids. Given a 3-D interpolation from a source 3-D grid (expressed as $H2D_s + V1D_s$) to a target grid
($H2D_t + V1D_t$), the 2-D interpolation between the horizontal sub grids $H2D_s$ and $H2D_t$ eventually interpolates coupling fields from the source 3-D grid to an intermediate 3-D grid consisting of $H2D_t$ and $V1D_s$, and thus the 1-D interpolation between the vertical sub grids $V1D_s$ and $V1D_t$ eventually interpolates coupling fields from the intermediate 3-D grid to the target 3-D grid. Specifically, the 2-D interpolation will be performed a number of times, each of which corresponds to a horizontal level of the source and intermediate 3-D grids, and the 1-D interpolation will also be performed for a number of times, each
corresponding to a column in the intermediate and target 3-D grids. For dynamic 3-D coupling, 2-D interpolation can use pre-calculated remapping weights, because the horizontal sub grids do not change throughout a simulation, while 1-D interpolation cannot use pre-calculated remapping weights, and instead must dynamically calculate the remapping weights according to the changes of vertical coordinate values in the source or target 3-D grid in time integration. To achieve dynamic 3-D interpolation based on the implementation of static 3-D interpolation in C-Coupler1, dynamic calculation for 1-
D remapping weights is implemented with the following steps in C-Coupler2.

1) If the source 3-D grid has a variable surface field, the import interface first receives the source surface field transferred from the export interface, and next uses the pre-calculated horizontal remapping weights to interpolate the source surface field from the source horizontal grid (the horizontal sub grid of the source 3-D grid) to the target horizontal grid



(data interpolation will be bypassed if the two horizontal grids are the same). The source surface field on the target horizontal grid will be used as the surface field of the intermediate 3-D grid, and will be further used as the target surface field when the target 3-D grid has an external surface field.

2) If the source 3-D grid has an external surface field, the import interface uses the target surface field as the surface field
of the intermediate 3-D grid (the target 3-D grid must have a non-external surface field in this case).

3) The import interface calculates the vertical coordinate values of the intermediate/target 3-D grid when the 3-D grid has a surface field (the import interface can obtain all constant information of the source 3-D grid in coupling generation before the first execution of the corresponding export and import interfaces).

4) For each column in the intermediate or target 3-D grid, the import interface calculates the 1-D remapping weights.

As dynamic 3-D interpolation cannot fully utilize pre-calculated remapping weights and must update 1-D remapping weights at almost all coupling steps, it has a higher computational cost than static 3-D interpolation. To minimize the impact of the increased computation cost, all of the above steps, including data transfer for the source surface field, 2-D interpolation for the source surface field, calculation of vertical coordinates of the intermediate/target 3-D grid, and calculation of 1-D remapping weights, are parallelized based on the MPI processes and parallel decompositions in the
corresponding component models. Moreover, the implementations of static 3-D interpolation and dynamic 3-D interpolation are unified. In detail, static 3-D interpolation will be treated as dynamic 3-D interpolation at the first step of coupling; the 1-D remapping weights will thus be calculated online during the first step of coupling; and they will be treated as static 3-D interpolation and use the existing remapping weights in the subsequent coupling steps.

## 4.5    Non-blocking data transfer

Almost all existing couplers use two-sided MPI communication (e.g., *MPI_Send*, *MPI_Recv*, and their non-blocking mode *MPI_Isend*, *MPI_Irecv*) to transfer data. Non-blocking data transfer is a necessary function of OASIS3-MCT, because it also can achieve model coupling within one executable or a subset of MPI processes. It still uses two-sided MPI communication, where the non-blocking mode is used to make the puts of coupling fields (corresponding to the export interfaces of C-Coupler2) generally non-blocking. Unpredictable "deadlocks" in non-blocking two-sided MPI-
communication-based data transfer can occur when an excessive number of messages sent to a single MPI process exhaust the message passing buffer space allocated by the MPI library (Dennis et al., 2012). C-Coupler2 therefore foregoes two-sided MPI communication, and instead uses one-sided MPI communication (i.e., *MPI_Put* and *MPI_Get*) for data transfer in a non-blocking manner, thus enabling the coupler to manage the message passing buffer space, while ensuring "safe" implementation of non-blocking data transfer.

Although non-blocking data send and receive operations are not blocked immediately, a mechanism that later blocks non-blocking data transfer is still necessary to guarantee successful and correct completion of each data transfer. In OASIS3-MCT, gets of coupling fields (corresponding to the import interfaces of C-Coupler2) are blocking, and a put will wait for the



completion of the last put of the same coupling field. Two-sided MPI communication is easily achieved through the MPI function *MPI_Wait*. However, there is no corresponding MPI function available for one-sided MPI communication, and thus extra effort is required to implement such a mechanism. In detail, non-blocking data transfer based on one-sided MPI communication is implemented as follows in C-Coupler2.

5  1)  The export interface calls *MPI_Put* to send coupling fields to the message passing buffer managed by the import interface, while the import interface obtains coupling fields from its message passing buffer.

2)  Before sending coupling fields, the export interface examines whether the message passing buffer of the import interface is available. The message passing buffer remains unavailable until the import interface has received coupling fields from the last data transfer. Before obtaining coupling fields, the import interface checks whether the data in the message passing buffer have been fully updated. After obtaining coupling fields, the message passing buffer is set as available. A time tag identifies the status of the message passing buffer (i.e., available or unavailable and data fully updated or not). The export interface uses *MPI_Get*/*MPI_Put* to query/update the status of the message passing buffer.

3)  Similarly to OASIS3-MCT, the data receive command issued by the C-Coupler2 API calls of executing import interfaces is blocking. Beyond the API calls, C-Coupler2 issues additional non-blocking data receives for local import interfaces, in order to make the data receive finish and the data send execute as early as possible. In a non-blocking data receive, C-Coupler2 will obtain the coupling fields and set the message passing buffer as available if the buffer has been fully updated; otherwise, it will do nothing.

As noted above, C-Coupler2 provides flexibility in setting a lag on a coupling connection via the "*remote_lag_count*" in the timer of the import interface. One challenge associated with this function is that a deadlock can occur if the lag is greater than the corresponding coupling period. For example, Fig. 10 includes two component models (*comp1* and *comp2*) coupled with two connections. The first coupling connection is from the export interface *exp1* of *comp1* to the import interface *imp2* of *comp2*, while the second is from the export interface *exp2* of *comp2* to the import interface *imp1* of *comp1*. Both connections have equal coupling periods of 600 s. At each coupling step of each model, the export interface is executed before the import interface. In Fig. 10(a), there is no lag on each coupling connection, and thus *comp1* and *comp2* can run concurrently. In Fig. 10(b), there is no lag on the second coupling connection, while the first coupling connection has a lag of 600 s, which means that *imp2* at the current coupling step wants the coupling fields from *exp1* at the previous coupling step. At the first coupling step, *exp1_1* (meaning *exp1* executed at the first coupling step) tries to send coupling fields to *imp2_2*. As no-blocking data transfer is used, *exp1_1* can successfully put the coupling fields into the message passing buffer of *imp2*, and thus *comp1* can finish *imp1_1*, and so finish the first coupling step. At the same time, *comp2* can finish the first coupling step (it is unnecessary to execute *imp2_1*). At the second coupling step, *exp1_2* will first await the message passing buffer of *imp2* that still keeps the coupling fields from *exp1_1*. After *comp2* finishes *exp2_2*, *imp2_2* is executed, and the message passing buffer of *imp2* will be set as available. Next, *exp1_2* can successfully put the coupling fields into the message passing buffer of *imp2*. Although the lag of 600 s on the second coupling connection does not introduce a deadlock, a problem in the sequencing of *comp1* and *comp2* is introduced: *exp1* in *comp1* must await the completion of *imp2* in *comp2*




at the same coupling step. In Fig. 10(c), there is no lag on the second coupling connection, while the first has a lag of 1200 s. Similarly to Fig. 10(b), both *comp1* and *comp2* can finish the first coupling step. At the second coupling step, *exp1_2* first awaits the message passing buffer of *imp2* that will not be set as available before *comp2* finishes *imp2_3* at the third coupling step, while *exp2_2* can successfully put the coupling fields into the message passing buffer of *imp1*, which has been set as available by *imp1_1* at the first coupling step. At the third coupling step of *comp2* (*exp1_2* in *comp1* is still waiting at the second coupling step), *exp2_3* first awaits the message passing buffer of *imp1* that will not be set as available before *comp1* finishes *imp1_2*. As a result, both *comp1* and *comp2* wait for each other, causing a deadlock.

The deadlock in Fig. 10(c) is not unbreakable, as it results from the unavailability of message passing buffers, and it can be avoided by increasing the message passing buffers of the corresponding import interfaces. In C-Coupler2, the message passing buffers of an import interface can be increased adaptively. At each time step of a component model, C-Coupler2 checks each import interface and will increase the message passing buffers of an import interface when all existing message passing buffers are unavailable. As a result, a positive lag on a coupling connection will never result in a deadlock or sequencing problem between component models.

A negative lag can also be specified for a coupling connection, but can result in sequencing problems between component models or even an unbreakable deadlock. For example, Fig. 10(d) shows no lag on the second coupling connection and a lag of −600 s on the first, which means that *imp2* at the current coupling step wants the coupling fields from *exp1* at the next coupling step. This lag setting will not introduce a deadlock, but will introduce a sequencing problem between *comp1* and *comp2*: *imp1* is coupled with *exp2* at the same step, while *imp2* at the current coupling step waits for *exp1* at the next coupling step. In Fig. 10(e), there is no lag on the second coupling connection, while the first has a lag of −1200 s, which introduces an unbreakable deadlock corresponding to the red arrows in the figure, where import interfaces are awaiting the export interfaces that cannot be executed until the import interfaces return.



### 4.6    Facilitation for model nesting

For a regional model without self-nesting capability (i.e., it can only manage a unique grid domain), C-Coupler2 can it help achieve self-nesting capability as follows.

1)  The code of the regional model can still only manage a unique grid domain, but multiple grid domains for self-nesting can be achieved through running multiple copies of the executable of the regional model, each of which can have separate input parameters and input data files for a unique grid domain and can be registered as a separate component model of C-Coupler2. The different grid domains should have different component model names, but they can use the same names for the model grids, parallel decompositions, coupling fields, coupling interfaces, etc. Therefore, C-Coupler2 only requires the regional model to obtain a few additional input parameters. In other words, slight modification of the *namelist* file and the corresponding model code of the regional model can enable C-Coupler2 to recognize multiple grid domains.

2)  Given that a small grid domain is nested in a larger grid domain, C-Coupler2 can recognize the relationship between the two grid domains through the coupling connection configuration files. As all grid domains can correspond to identical code in the regional model, the coupling connection configuration files of different grid domains can be similar, differing only in terms of the full names of component models in the file contents. Thus, the coupling connection configuration files of all grid domains can be generated easily or even automatically by a script.

3)  Self-nesting capability requires the exchange of model fields that are generally 3-D between grid domains. Implementation of this exchanging can benefit from the 3-D coupling capability, especially the dynamic 3-D coupling capability, of C-Coupler2. Moreover, given that a small grid domain is nested in a larger grid domain, the coupling procedures for exchanging model fields between them can be automatically generated in their partial coupling generation.

C-Coupler2 does not provide any lateral boundary condition scheme. This is not a problem, because a regional model generally includes lateral boundary condition schemes that can also be used in self-nesting. To achieve two-way self-nesting, schemes for using the feedback from smaller grid domains should be added to the regional model.

As each grid domain corresponds to a separate copy of the executable, each grid domain can easily use a distinct subset of MPI processes, which allows simultaneous integration of grid domains for better parallel performance. Scientists may want to integrate a grid domain earlier than its nested grid domains. For example, after a grid domain finishes integration



from 0 to 90 s, its nested grid domains can start the integration from 0 s. This can be achieved by adjusting coupling latencies among grid domains based on the timers of the corresponding import interfaces. In one-way nesting, the coupling latencies generally do not affect the parallelism among grid domains. In other words, different grid domains can always be integrated simultaneously in one-way nesting, regardless of the coupling latencies. In two-way nesting, even when a specific

setting of coupling latencies forces sequential running between a grid domain and its nested grid domains, multiple nested grid domains of the same grid domain can also run simultaneously, so that C-Coupler can also help improve the parallel performance of self-nesting. One challenge resulting from sequential running is that the corresponding processors will be essentially idle, and therefore wasted, when a grid domain is waiting for another grid domain. In the future, we will investigate technical solutions to overcome this inefficiency. Moreover, it may be an interesting topic to investigate the

scientific impact of different settings of coupling latencies in model nesting.

Similarly, it can also benefit from C-Coupler2 to nest a regional model into a different model. For a regional model that already has self-nesting capability (such as WRF), C-Coupler2 can identify each grid domain as a component model. Therefore, a grid domain in self-nesting can be further coupled with another kind of component model.

### 4.7    Facilitation for incremental coupling

Incremental coupling can be viewed as coupling external component models with an existing coupled model. A straightforward implementation is to treat the external component models as internal component models of the existing coupled model, and use the coupler of the existing coupled model to handle the corresponding incremental coupling. For example, regarding the work of nesting WRF into CESM done by He et al. (2013) that has been introduced in Section 3.7, WRF is treated as an internal component model of CESM, and the incremental coupling for its nesting is handled by CPL7,

the coupler of CESM. A major challenge in this kind of implementation is that the independence might need to be broken between external component models and the existing coupled model that may have been developed independently by different groups for a number of years. This introduces significant code changes to the models (even including the coupler), and results in inconsistent code versions of the same model among different model groups. For example, all component models of CESM share the same driver and are compiled into a unique executable, while WRF has its own driver, different

from the others. When treating WRF as an internal component model of CESM, WRF will have to use the driver of CESM, and will also be compiled into the unique executable. Thus, WRF's original driver and compiling scripts as well as CESM need to be modified. Moreover, as the original driver and coupler of CESM do not consider the existence of a regional atmosphere model, the driver and coupler codes of CESM also need to be modified.

Incremental coupling faces the fundamental problem of guaranteeing independence between external component

models and the existing coupled model, so as to minimize code changes to the models or the coupler. To help in this regard, C-Coupler2 should minimize the constraints on using external component models and existing coupled models that are already coupled with other couples; it should also work as an additional coupler specifically for incremental coupling as part





of coupling in a new coupled model, thus letting developers focus only on the coupling between external component models and the corresponding component models in the existing coupled model. In response to these requirements, C-Coupler2 includes the following implementations for incremental coupling.

1) An existing coupled model can be registered to C-Coupler2 as a component model, and its component models involved
in incremental coupling can be further registered as its children, while other component models that are irrelevant to incremental coupling can be neglected. Generally, several API calls are enough for the model registration, which only introduces slight code changes to the existing coupled model. As C-Coupler2 can support almost any MPI process layout among component models, a component model in any existing coupled model can be easily registered to C-Coupler2. Similarly, an external component model can be easily registered to C-Coupler2.

2) As C-Coupler2 allows coupling generation to be performed at any time for any subset of component models, partial coupling generations for only the component models relevant to incremental coupling can be performed flexibly. Similarly, several API calls are enough for partial coupling generations, which only introduces a slight code change to the external component models and the existing coupled model.

## 4.8 Debugging capability

The following aspects enhance the debugging capability of C-Coupler2.

1) C-Coupler2 performs a series of checks for almost all API calls. For example, when registering a component model, model grid, parallel decomposition, coupling field, or coupling interface, when setting the time step of a component model, and when executing a coupling interface, C-Coupler2 can check whether all MPI processes of the component model call the API at the same time and with consistent parameters. For example, when registering a horizontal grid
with global grid data or registering a vertical grid, C-Coupler2 can check whether the grid data are the same among MPI processes, and when registering a coupling interface, C-Coupler2 can check whether the timer, coupling field instances, and other parameters are consistent among MPI processes. When an API call includes an array as a parameter, C-Coupler2 can check the size of the array. For example, when registering a coupling field instance, C-Coupler2 can check whether the array size of the memory buffer of the coupling field instance matches that required. When an API call
includes the ID of a coupling resource as a parameter, C-Coupler2 can check whether the ID is legal. When an API call will read information in configuration files, C-Coupler2 can check whether the files are in the correct XML format and check the correctness of the required information. Given the additional overheads in computation and communication introduced by performing such checks, most of them can be disabled in a model run. We strongly recommend that the user enables the checks fully when developing a coupled model.

2) When an error or a warning is detected, it will be reported, including a suggestion for fixing the relevant model codes or configuration files. Almost all APIs include an optional input parameter "annotation", which is a string giving a hint for locating the model code of the API call corresponding to an error or warning. There are around 1000 error reports

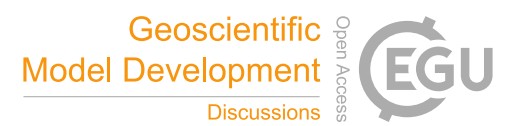

throughout the code of C-Coupler2.

3) C-Coupler2 can report many kinds of log information, about coupling configurations, progress in handling coupling configurations, coupling generations, coupling run, and the values of coupling field instances. Each process in a coupled model can have a separate log file for reporting log information, errors, and warnings, which can facilitate parallel debugging. Moreover, C-Coupler2 also enables the internal code of a component model to report log information through the C-Coupler2 log files. The user can enable or disable the reporting of log information.

## 5   Evaluation

This section evaluates C-Coupler2 in several aspects, including software testing, scaling of initialization, data transfer, and memory use.

### 5.1   Software testing

To improve the reliability of C-Coupler2 in various areas of application, we first designed a sample coupled model that includes coupling between several sample component models and self-nesting component models. Next, we developed hundreds of use cases based on the sample coupled model, to evaluate whether C-Coupler2 properly detects and reports errors in various cases of incorrect use, and properly generates coupling procedures and handles model coupling and nesting in correct cases.

Besides the sample coupled model, existing real coupled models were used to test C-Coupler2, including FIO-AOW, BCC_CSM (Xin et al., 2013), and CESM. For each coupled model, we used C-Coupler2 to replace some coupling functions from other couplers, while trying to keep exactly the same (bitwise identical) simulation results. Specifically, we used C-Coupler2 to replace C-Coupler1 employed in FIO-AOW, MCT (Larson et al., 2005) based data transfer and interpolation functions of the CPL5 coupler (the version immediately prior to CPL6) employed in BCC_CSM, and MCT based data transfer and interpolation functions of the CPL7 coupler employed in CESM. Various MPI process numbers of component models and various run types (i.e., "*initial*" "*continue*", "*branch*", and "*hybrid*" run) of each real coupled model were used for testing using nearly 2000 processor cores. As CESM enables the user to flexibly set MPI process layouts among component models, the tests considered non-overlapping, partially overlapping, and overlapping MPI processes among component models of CESM.

We further coupled an atmosphere model GAMIL (Li et al., 2013) and the CESM version with C-Coupler2 (the original atmosphere model was disabled) via incremental coupling, which generated the executables of both GAMIL and CESM, while GAMIL kept its original driver unchanged. C-Coupler2 thus successfully demonstrated incremental coupling. The dynamic 3-D coupling capability of C-Coupler2 was evaluated when coupling GAMIL and GEOS-Chem.

Moreover, various processors (i.e., Intel X86, IBM Power, and the SW26010 processors employed in the Sunway TaihuLight system), various operating systems (i.e., Linux and IBM AIX), various compilers (i.e., Intel compilers, GNU





compilers and IBM compilers), and various MPI libraries (Intel MPI, MPICH, Open MPI, and IBM MPI) were involved in testing C-Coupler2.

## 5.2  Initialization cost

The evaluation of initialization cost considered coupling 10 fields between two toy component models that define horizontal grids but do not have real model initialization. The component models' horizontal grids were a regular longitude–latitude grid with $1440 \times 720$ grid points and a tripolar grid with $1440 \times 1021$ grid points. Therefore, model coupling required data remapping, and we used the corresponding remapping weights file generated by CoR1. The two component models ran concurrently on a supercomputer, with the same number of processor cores (MPI processes). Each computing node on the supercomputer included two Intel Xeon CPUs, with 20 processor cores in total, and all computing nodes were connected with an InfiniBand network. The codes were compiled by an Intel Fortran and C++ compiler at the optimization level O2, using an Intel MPI library.

Figure 11 shows the initialization cost of C-Coupler2 when scaling the number of processor cores of each component model from 15 to 960. It increases with increasing core number. C-Coupler2 initialization consists of several steps, including registering component models, setting time steps, registering model grids, registering parallel decompositions, registering coupling field instances, registering coupling interfaces, generating coupling procedures, reading and then distributing the remapping weights from a file, and computing routing information for data transfer between component models. Most of these steps include non-scalable operations; i.e., MPI collective communications or I/O. Increasing the core number increases the overhead of the non-scalable operations, and thus increases initialization cost. The results in Fig. 11 indicate that the initialization cost of C-Coupler2 may be not negligible, but it would be affordable in most cases, especially for long-term climate simulations.

## 5.3  Data transfer

As introduced above, C-Coupler2 employs asynchronous data transfer via one-sided MPI communication, while other existing couplers use two-sided MPI communication to transfer data. We evaluated the performance of our one-sided communication in comparison to a two-sided implementation, based on a ping-pong coupling for the same configuration used in Section 5.2. Figure 12 shows the cost of the two implementations when scaling the number of processor cores of each component model from 15 to 960. The times are per 100 ping-pong couplings. Overall, the one-sided communication achieves similar performance to the two-sided communication used in C-Coupler1. In other words, compared with C-Coupler1, C-Coupler2 achieves non-blocking data transfer through one-sided MPI communication without degrading the performance of data transfer.



## 5.4 Memory usage

Figure 13 shows the memory use per core for the configuration used in Section 5.2, as measured using the gptl (http://jmrosinski.github.io/GPTL/) interface (similar to Craig et al., 2017). The memory usage remained around 360–380 MB regardless of core number. Lacking computing resources, we were unable to evaluate the memory usage at a much

higher number of processor cores. However, we can speculate based on the assessment of OASIS3-MCT (Craig et al., 2017): owing to the MPI memory footprint (Balaji et al., 2008), the memory use per core might be around 1300–1400 MB at 16000 cores for each component model. The memory use is relatively high, but would be acceptable for many applications and hardware configurations.

## 6    Discussion and conclusion

As a new version of C-Coupler, C-Coupler2 follows the family's targets and the main designs, but is significantly different from C-Coupler1 in many aspects, as summarized in Table 10. Here we further discuss its capability in integrating external coupling algorithms. In C-Coupler1, a private subroutine of a component model or a common algorithm such as a flux calculation algorithm can be registered as an external coupling algorithm. An external coupling algorithm cannot have any explicit argument while its inputs and outputs are implicitly specified through the corresponding configuration files. The

integration of a Fortran external coupling algorithm generally requires an additional C interface. An external coupling algorithm can be further used as a runtime algorithm in a coupling procedure by specifying it in the corresponding configuration files. C-Coupler2 does not inherit this capability from C-Coupler1, because configuration files for implicitly specifying the inputs and outputs of an external coupling algorithm and for specifying the runtime algorithms in a coupling procedure do not exist in C-Coupler2. We intend to recover this capability in future versions of C-Coupler. As C-Coupler1's

coupling configuration interface has been significantly changed in C-Coupler2, C-Coupler2 does not achieve backwards compatibility. However, we will guarantee backwards compatibility in future C-Coupler versions.

3-D coupling capability is still a focus of C-Coupler development, and the static 3-D coupling capability in C-Coupler1 has been upgraded to dynamic 3-D coupling capability in C-Coupler2. Conserving model coupling is a fundamental requirement of a coupler, as it is critical to the stability of a long-time simulation of a coupled model. Most existing couplers

can achieve conservative 2-D coupling based on a conservative remapping algorithm on horizontal grids. C-Coupler2 does not guarantee conservation in 3-D coupling, while we are not aware of any 3-D conservative remapping algorithm available for model coupling. Previous works have demonstrated that it is practical to develop a common horizontal conservative remapping algorithm. In our opinion, the most significant reason is that a common approach, which can be described as area mapping, can guarantee conservation of coupling between horizontal grids. We are not sure whether a volume-mapping-

based approach can be used to develop a common 3-D conservative remapping algorithm, because a component model that can achieve 3-D conservation in time integration may have its own specific way to diagnose 3-D conservation, which is





generally determined by the dynamic core and may be not a volume-based approach. Moreover, it will be much more difficult to calculate volume mapping between 3-D grids, compared with the calculation of area mapping. Our future development of C-Coupler will involve investigations of 3-D conservative remapping schemes.

Although the results in Section 5 indicate that the initialization cost and memory use of C-Coupler2 may be affordable in most cases, a problem might arise when the model resolution or the number of processor cores is extremely high. When developing future C-Coupler versions, we will investigate ways to decrease initialization cost and memory use.

*Code availability.* The C-Coupler2 source code will be publicly available (e.g., through GitLab, GitHub or another public repository) no later than June 2018, and can be obtained from the authors on request before its publication.

*Acknowledgements.* This work was jointly supported in part by the National Grand Fundamental Research 973 Program of China (grant no. 2014CB441302) and the National Key Research Project of China (grant no. 2017YFC1501903).

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



Table 1  APIs for component model management

| No. | API | Brief description |
|---|---|---|
| 1 | CCPL_register_component | Register a component model to C-Coupler2 |
| 2 | CCPL_end_coupling_configuration | Finalize the coupling configuration stage of the given component model |
| 3 | CCPL_finalize | Finalize the model coupling by C-Coupler2 |
| 4 | CCPL_get_component_id | Get the ID of a component model corresponding to the given model name |
| 5 | CCPL_get_current_process_id_in_component | Get the ID of the current process in the given component model |
| 6 | CCPL_get_component_process_global_id | Get the global ID of a process in the given component model |
| 7 | CCPL_get_num_process_in_component | Get the total number of processes in the MPI communicator of the given component model |
| 8 | CCPL_get_local_comp_full_name | Get the full name of the given component model |
| 9 | CCPL_is_current_process_in_component | Check whether the current process is in the given component model |
| 10 | CCPL_is_comp_type_coupled | Check whether at least one component model with the given model type has already been registered to C-Coupler |



Table 2  Model types currently supported by C-Coupler2

| Model type | Description | Remark |
| --- | --- | --- |
| cpl | Coupler | Active component model |
| atm | Atmosphere model | Active component model |
| glc | Glacier model | Active component model |
| atm_chem | Atmospheric chemistry model | Active component model |
| ocn | Ocean model | Active component model |
| lnd | Land surface model | Active component model |
| sea_ice | Sea ice model | Active component model |
| wave | Wave model | Active component model |
| roff | Runoff model | Active component model |
| active_coupled_system | Coupled model that consists of a set of component models | Active component model |
| pseudo_coupled_system | Coupled model that consists of a set of component models | Pseudo component model |



Table 3  APIs for time management

| No. | API | Brief description |
| --- | --- | --- |
| 1 | CCPL_set_time_step | Set the unique time step of the given component model |
| 2 | CCPL_get_time_step | Get the time step of the given component model set by the API "CCPL_set_time_step" |
| 3 | CCPL_advance_time | Advance the model time of the given component model by a time step |
| 4 | CCPL_get_number_of_current_step | Get the number of the current time step of the given component model |
| 5 | CCPL_get_number_of_total_steps | Get the total number of time steps during the whole simulation of the given component model |
| 6 | CCPL_get_current_date | Get the current date of the given component model |
| 7 | CCPL_get_current_year | Get the current year of the given component model |
| 8 | CCPL_get_current_num_days_in_year | Get the current number of days in the current year of the given component model |
| 9 | CCPL_get_current_second | Get the current second of the given component model |
| 10 | CCPL_get_start_time | Get the start time of the model run of the given component model |
| 11 | CCPL_get_stop_time | Get the stop time of the model run of the given component model |
| 12 | CCPL_get_previous_time | Get the time of the previous time step of the given component model |
| 13 | CCPL_get_current_time | Get the current time of the given component model |
| 14 | CCPL_is_first_step | Check whether the current time step is the first step in a model run |
| 15 | CCPL_is_first_restart_step | Check whether the current time step is the first step after restarting the simulation run (first |





| | | restart step); the first restart step might not be the first step of the model run |
|---|---|---|
| 16 | CCPL_get_num_elapsed_days_from_start | Get the number of elapsed days from the start date to the current date of the given component model |
| 17 | CCPL_get_num_elapsed_days_from_reference | Get the number of elapsed days from the reference date to the current date of the given component model |
| 18 | CCPL_is_end_current_day | Check whether the current time is the end of the current day for the given component model |
| 19 | CCPL_is_end_current_month | Check whether the current time is the end of the current month for the given component model |
| 20 | CCPL_get_current_calendar_time | Get the current calendar time of the given component model |
| 21 | CCPL_check_current_time | Check the consistency in model time between the given component model and C-Coupler2 |
| 22 | CCPL_is_model_run_ended | Check whether the simulation run of the given component model has reached the stop time |
| 23 | CCPL_define_single_timer | Define a single, periodic timer for the given component model |
| 24 | CCPL_is_timer_on | Check whether the given timer is on at the current time step of the corresponding component model |
| 25 | CCPL_is_last_step_of_model_run | Check whether the current time step is the last time step of the model run of the given component model |
| 26 | CCPL_reset_current_time_to_start_time | Reset the current time of the given component model to the start time of the model run |





Table 4  APIs for grid management

| No. | API | Brief description |
| --- | --- | --- |
| 1 | CCPL_register_H2D_grid_via_global_data | Register a horizontal grid using global grid data |
| 2 | CCPL_register_H2D_grid_via_local_data | Register a horizontal grid using local grid data |
| 3 | CCPL_register_H2D_grid_via_file | Register a horizontal grid using a data file |
| 4 | CCPL_register_H2D_grid_from_another_component | Register a horizontal grid based on another component model |
| 5 | CCPL_register_V1D_Z_grid_via_model_data | Register a vertical Z grid using model data |
| 6 | CCPL_register_V1D_SIGMA_grid_via_model_data | Register a vertical SIGMA grid using model data |
| 7 | CCPL_register_V1D_HYBRID_grid_via_model_data | Register a vertical HYBRID grid using model data |
| 8 | CCPL_register_MD_grid_via_multi_grids | Register a grid using multiple registered grids |
| 9 | CCPL_register_mid_point_grid | Register the mid-point grid of a 3-D interface-level grid |
| 10 | CCPL_set_3D_grid_variable_surface_field | Set the dynamic surface field of a 3-D grid |
| 11 | CCPL_set_3D_grid_constant_surface_field | Set the static surface field of a 3-D grid |
| 12 | CCPL_set_3D_grid_external_surface_field | Declare that the surface field of a 3-D grid is external |
| 13 | CCPL_get_grid_size | Get the global size of a grid |
| 14 | CCPL_get_grid_id | Get the ID of a grid |
| 15 | CCPL_get_H2D_grid_data | Get grid data of a horizontal grid |



Table 5  API for parallel decomposition management

| No. | API | Brief description |
|-----|-----|-------------------|
| 1 | CCPL_register_parallel_decomp | Register a parallel decomposition on a horizontal grid of the given component model |



Table 6  API for coupling field instance management

| No. | API | Brief description |
| --- | --- | --- |
| 1 | CCPL_register_field_instance | Register a coupling field instance of the given component model |





Table 7  APIs for coupling interface management

| No. | API | Brief description |
|---|---|---|
| 1 | CCPL_register_export_interface | Register a coupling interface for exporting field instances |
| 2 | CCPL_register_import_interface | Register a coupling interface for importing field instances |
| 3 | CCPL_register_normal_remap_interface | Register a coupling interface for remapping field instances normally |
| 4 | CCPL_register_frac_based_remap_interface | Register a coupling interface for remapping field instances where fractions will be used |
| 5 | CCPL_execute_interface_using_id | Execute a coupling interface specified by an interface ID |
| 6 | CCPL_execute_interface_using_name | Execute a coupling interface in a component model specified by an interface name |
| 7 | CCPL_get_H2D_grid_area_in_remapping_wgts | Get the area of grid cells of a horizontal grid that are calculated in remapping weight calculations in coupling generation for the given coupling field instance of the given coupling interface |
| 8 | CCPL_check_is_import_field_connected | Check whether the given import field instance in the given import interface has already been connected through coupling generation |





Table 8  APIs for coupling generation

| No. | API | Brief description |
| --- | --- | --- |
| 1 | CCPL_do_individual_coupling_generation | Do coupling generation within the given component model |
| 2 | CCPL_do_family_coupling_generation | Do family coupling generation among the given component model and its descendant component models that are not disabled in family coupling generation |
| 3 | CCPL_do_external_coupling_generation | Do coupling generation for a given set of component models; either individual or family coupling generation can be specified for each given component model |
| 4 | CCPL_get_configurable_comps_full_names | Get the full names (and individual or family coupling generation specification) of a set of component models from the corresponding XML configuration file |





Table 9  APIs for restart management and parallel debugging

| No. | API | Brief description |
|---|---|---|
| 1 | CCPL_do_restart_write_IO | Write local variables or data related to C-Coupler2 into data file |
| 2 | CCPL_do_restart_read_IO | Read in local variables or data related to C-Coupler2 from a restart data file |
| 3 | CCPL_is_restart_timer_on | Check whether the implicit restart timer is on |
| 4 | CCPL_report_log | Write a log for a given component model into a corresponding log file managed by C-Coupler2; each process of each component model can write log information into a separate log file |
| 5 | CCPL_report_progress | Write a progress report for a given component model into a corresponding log file managed by C-Coupler2; only the root process of a component model can write a progress report |
| 6 | CCPL_report_error | Write an error report for a given component model into a corresponding log file managed by C-Coupler2, and then stop the whole model run; each process of each component model can write an error report into a separate log file |
| 7 | CCPL_get_comp_log_file_name | Get the log file name of the current process of the given component model; this log file is not used by C-Coupler2 (i.e., the coupler will not write any report into it); a component model can further open the corresponding log file, and then write its log information into it |
| 8 | CCPL_get_comp_log_file_device | Get the device ID of the log file of the current process of the given component |





| | | model; if the log file has not been opened, it will be opened to a device ID; this log file is not used by C-Coupler2 (i.e., the coupler will not write any report into it); a component model can further use the device ID to write its log information into the corresponding log file |
|---|---|---|





Table 10  Differences between C-Coupler1 and C-Coupler2

|  | C-Coupler1 | C-Coupler2 |
|---|---|---|
| Coupling configuration | Strongly dependent on ASCII formatted configuration files | Properly combines APIs and XML formatted configuration files |
| MPI process layout | Only root component models are supported; cannot handle model coupling within a subset of MPI processes or the same component model | Can support almost any kind of MPI process layout among component models (including existing coupled models); can handle model coupling within a subset of MPI processes or the same component model |
| 3-D coupling capability | Static 3-D coupling only | Both static and dynamic 3-D coupling |
| Coupling generation | No coupling generation function | Automatic and incremental coupling generation for any subset of component models at any time |
| Data transfer | Blocking data transfer based on *MPI_Send*/*MPI_Isend* and *MPI_Recv*/*MPI_Irecv* | Non-blocking data transfer based on *MPI_Put*/*MPI_Get* |
| Support for model nesting | No specific support | Can facilitate to nest a regional model into itself or into another model |
| Support for incremental coupling | No specific support | Can facilitate to couple external component models with an existing coupled model |
| Debugging capability | Not prioritized; little support | Implemented, with support provided |
| Coupling lags | Supported, but lag cannot be greater than the corresponding coupling period | Supported, with lag able to be greater than the corresponding coupling period |
| Coupling procedures | Runtime algorithms in a coupling procedure are explicitly specified in a configuration file | Runtime algorithms in a coupling procedure are implicitly generated by the coupling generator |
| Capability of integrating external coupling algorithms | Can integrate an external coupling algorithm as a runtime algorithm and then further use it in a coupling procedure | Cannot integrate an external coupling algorithm |





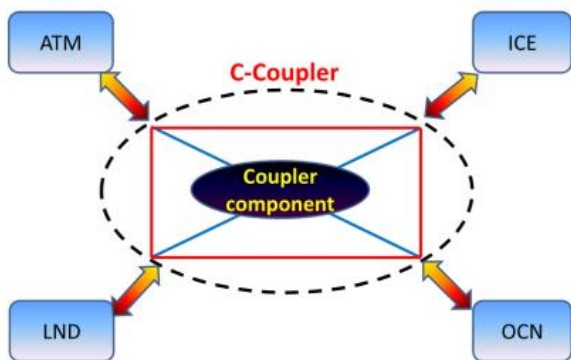

Figure 1 General architecture of models coupled with C-Coupler. "ATM" means an atmosphere model, "OCN" means an ocean model, "ICE" means a sea ice model and "LND" means a land surface model. (from Liu et al., 2014).





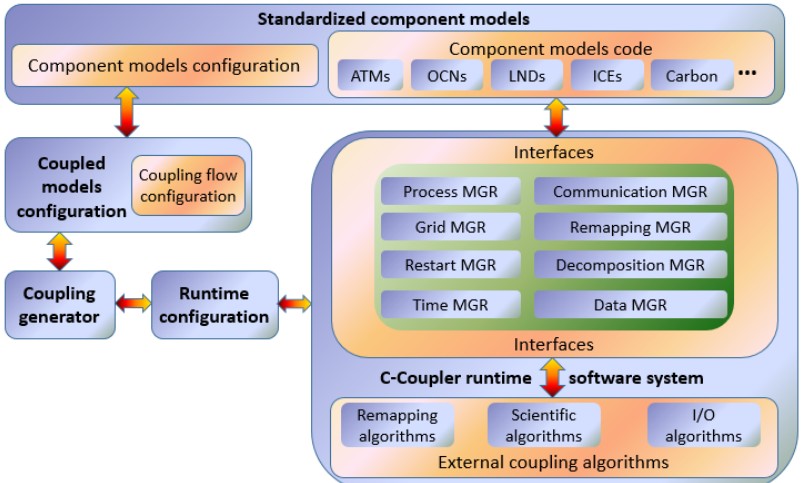

Figure 2  General software structure of C-Coupler (from Liu et al., 2014).





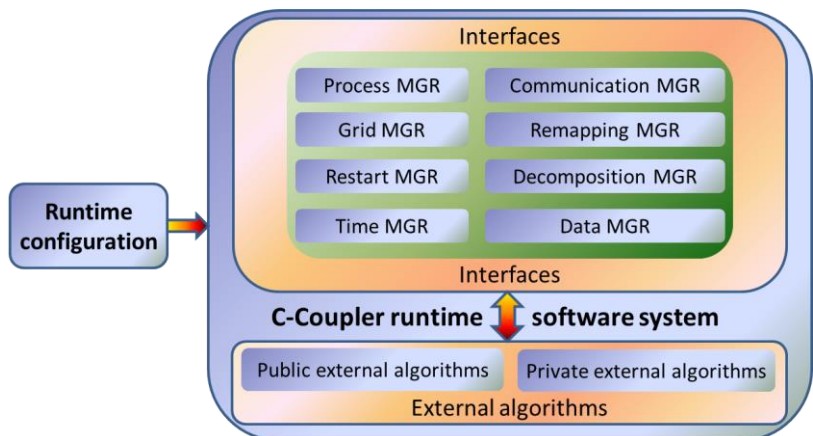

Figure 3  C-Coupler1 software structure.





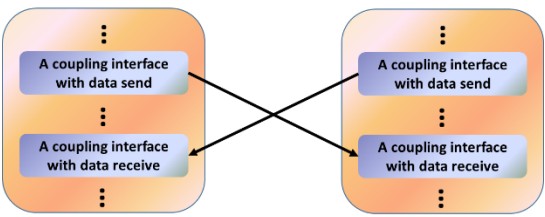

Figure 4  Example of model coupling between two component models: deadlocks occur when blocking data send/receive operations are used.





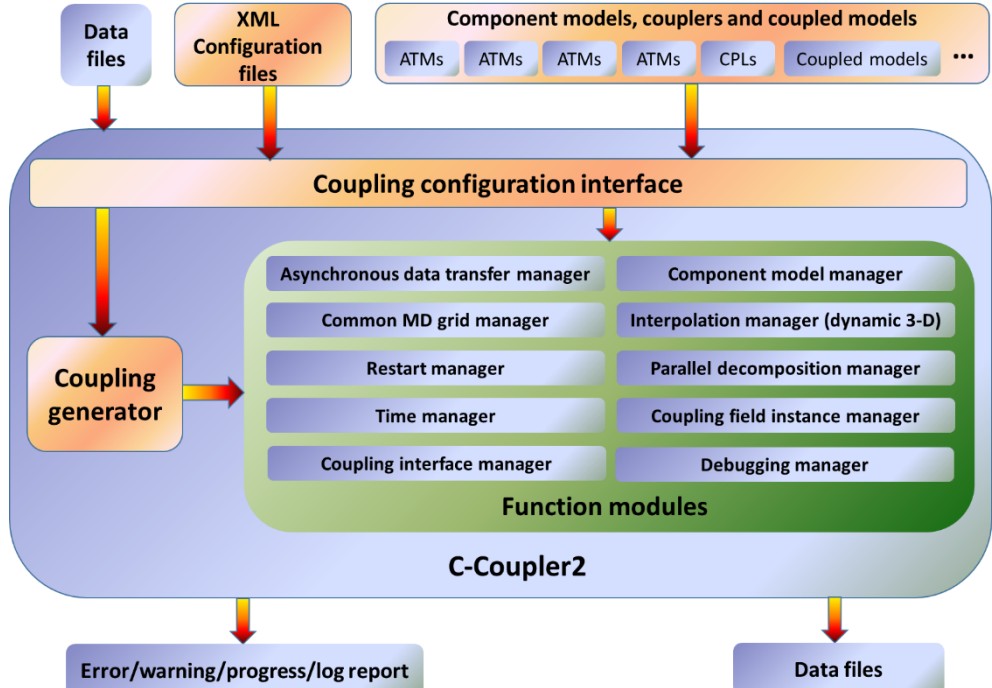

Figure 5  C-Coupler2 software structure.



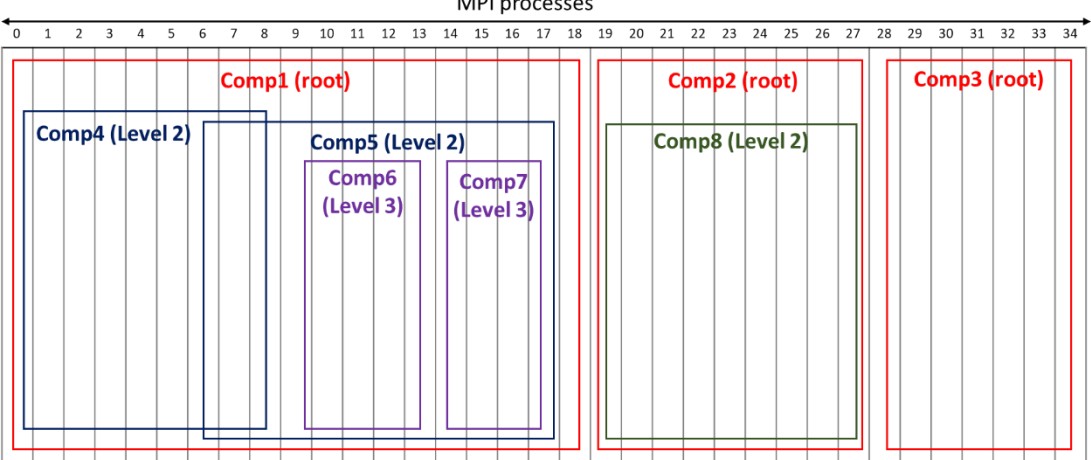

Figure 6  Sample process layout of component models (comp1–comp8).





Figure 7  An example of hybrid coupling configuration and model coupling in the initialization stage of a coupled model constructed with C-Coupler2. *Comp1–comp4* are the four component models. Labels and boxes of the same color correspond to the same component model.





Figure 8  An example of model coupling in the kernel (time integration) stage of a coupled model constructed with C-Coupler2. *Comp1–comp4* are the four component models. Labels and boxes of the same color correspond to the same component model.



```
       <root>
L1:    <remapping_setting status="on">
L2:        <remapping_algorithms status="on">
L3:          <H2D_algorithm status="on"  name="bilinear">
L4:            
L5:          </H2D_algorithm>
L6:          <V1D_algorithm status="on" name="linear">
L7:            
L8:          </V1D_algorithm>
L9:          <H2D_weights  status="on">
L10:           <file name="map_to_global_grid1_default.nc" />
L11:           <file name="map_to_regional_grid1_default.nc" />
L12:          </H2D_weights>
L13:        </remapping_algorithms>
L14:      <fields  status="on" specification="default" />
L15:   </remapping_setting>

L16:   <remapping_setting  status="on">
L17:        <remapping_algorithms status="on">
L18:          <H2D_algorithm  status="on"  name="conserv_2D" />
L19:          <H2D_weights  status="on">
L20:            <file name="map_to_global_grid1_conserv.nc" />
L21:            <file name="map_to_regional_grid1_conserv.nc" />
L22:          </H2D_weights>
L23:        </remapping_algorithms>
L24:      <fields  status="on" specification="type">
L25:        <entry value="flux" />
L26:      </fields>
L27:   </remapping_setting>
L28:   <remapping_setting  status="on">
L29:        <remapping_algorithms status="on">
L30:          <V1D_algorithm status="on" name="linear">
L31:            
L32:            
L33:          </V1D_algorithm>
L34:        </remapping_algorithms>
L35:      <fields  status="on" specification="name">
L36:        <entry value="t_atm_3D" />
L37:        <entry value="ghs_atm_3D" />
L38:      </fields>
L39:   </remapping_setting>
       </root>
```

Figure 9  Sample of a remapping configuration file





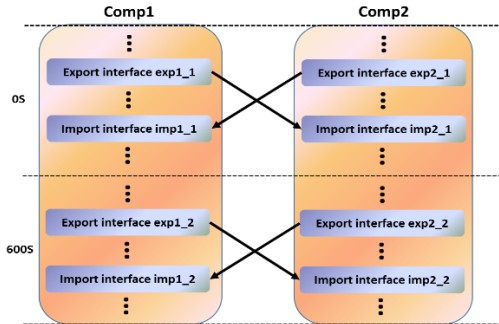

(a) Two-way coupling without lags

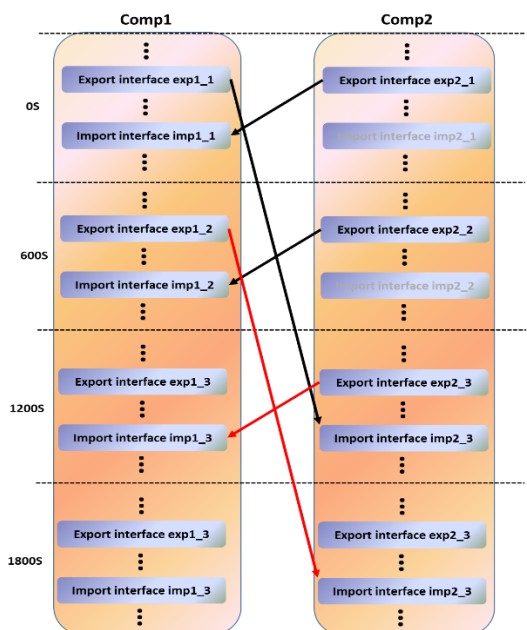

(c) Two-way coupling with a lag of 1200 s from *comp1* to *comp2*

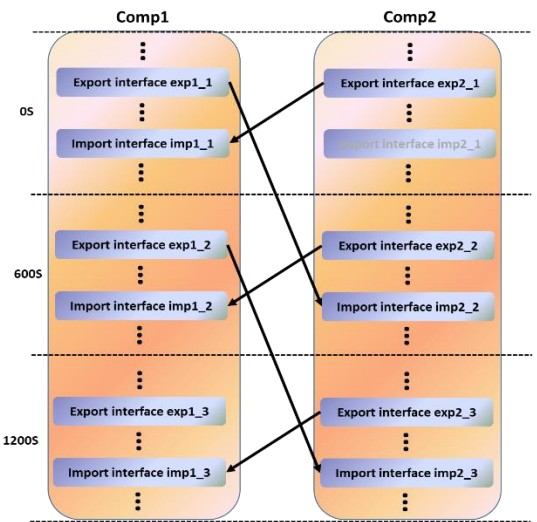

(b) Two-way coupling with a lag of 600 s from *comp1* to *comp2*





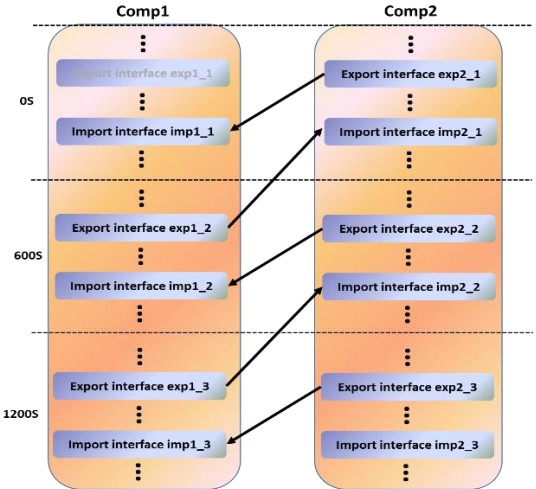
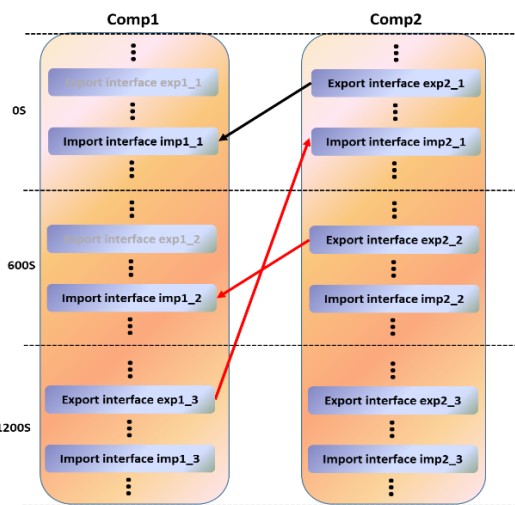

(d) Two-way coupling with a lag of −600 s from *comp1* to *comp2*

(e) Two-way coupling with a lag of −1200 s from *comp1* to *comp2*

Figure 10  Sample two-way couplings with different lag settings





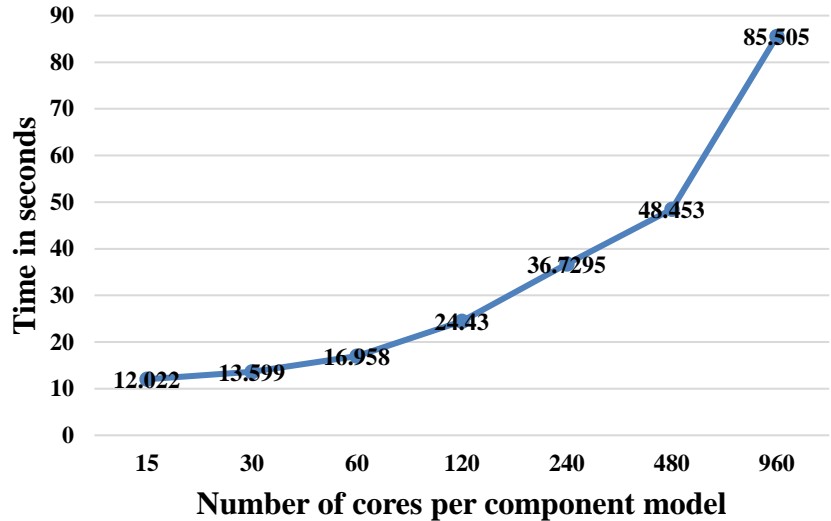

Figure 11  Initialization cost for coupling two toy models with C-Coupler2 on a supercomputer with Intel Xeon CPUs and an InfiniBand network.





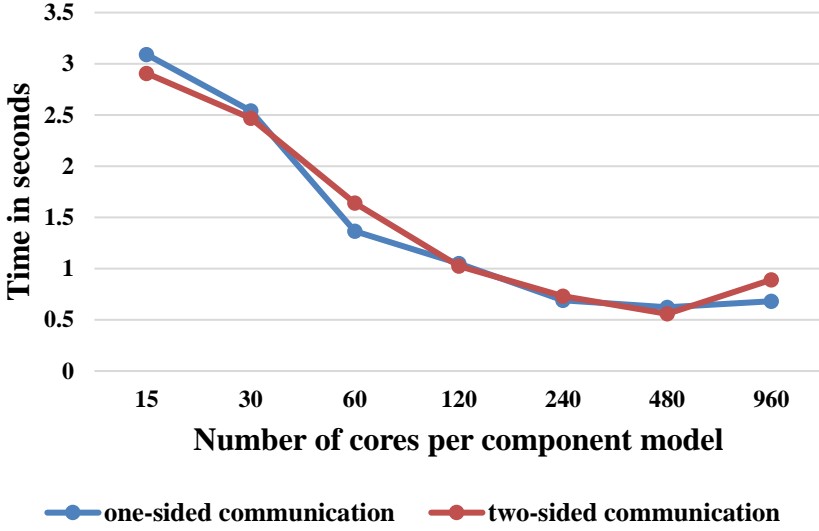

Figure 12  Comparison of data transfer times (for 100 ping-pong couplings) between a one-sided and a two-sided

implementation, with the same configuration as Fig. 11.





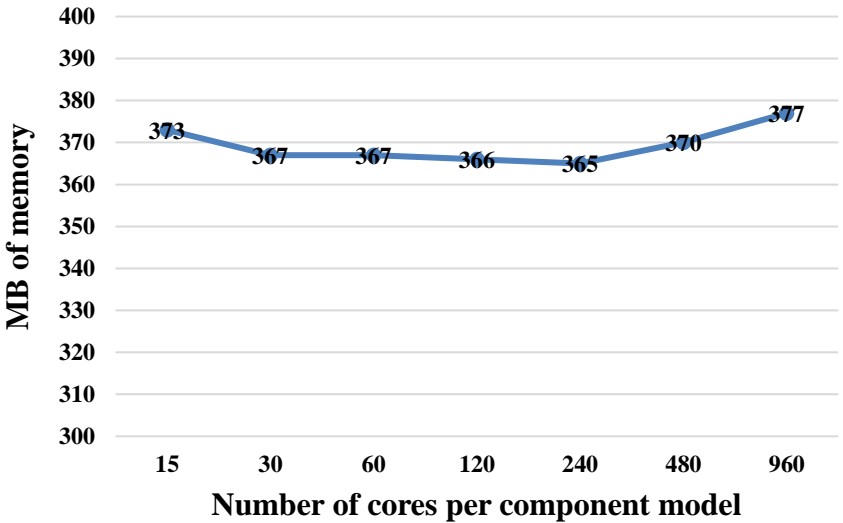

Figure 13  Memory use of C-Coupler2 for the toy coupled model considered in Section 5.2.