# Peer review of "C-Coupler2: a flexible and user-friendly community coupler for model coupling and nesting"

_Geoscientific Model Development, 2018_

## Referee Comment (RC1) · Anonymous Referee #1 · 9 Mar 2018

Title: C-Coupler2: a flexible and user-friendly community coupler for model coupling and nesting Author(s): Li Liu et al. MS No.: gmd-2018-27

General Comments:

This paper presents an overview of the updated C-Coupler, C-Coupler2. Overall, it is well written and provides both the motivation and implementation details for the up- dated version of the model. This paper is primarily am overview paper and it does not contain much application or performance results, but I feel this is reasonable at this time. I would encourage the authors to publish additional results in the future detailing the performance cost of higher resolution and higher core counts tests and sharing the performance of 3D weight generation and coupling.

[Figure]

Specific Comments:

page 23, lines 25-29. The ability to run with lags properly is critical. Lags almost always create additional requirements on restart files and the ability to restart a model exactly (bit-for-bit) with lagged coupling fields should be a requirement if lags are going to be supported. It sounds like this is not currently supported in C-Coupler2? Maybe rather than saying "We therefore propose", it would be clearer to say something like "Lags are not fully supported in the current version ,but in the future, the C-Coupler2 will . . ."

page 26, Section 4.5. Is the only reason MPI_Put and MPI_Get is used is to avoid possible exhaustion of MPI buffer space? That should be very rare in practice. Are there other reasons? Performance, ease of implementation, etc? Based on the description on Section 4.5, the MPI_Put/Get implementation sounds slower and more complicated than well managed MPI_ISend/IRecv implementations with MPI_Wait implemented appropriately. Are the authors happy with this implementation? Section 5.3 answers this question in part, but it might be nice to add a few more words in either section 4.5 or 5.3. I think one-side communication potentially helps with both MPI buffer usage and ability to have greater flexibility in coupling lags, but does not improve performance? How about implementation complexity?

page 28, line 8. This is a nice feature. One has to be concerned about memory usage but this provides a nice way to allow extra flexibility in lags compared to other implementations.

page 22, paragraph beginning at line 6. I believe the comparison between Oasis3-MCT_3.0, CESM, and C-Coupler2 is not particularly clear. The authors compare how components interact in different systems, but the definition of the component is not the same in each system. In CESM and the C-Coupler2, the component is defined by separation of scientific models. In Oasis, the component is defined by the separation of MPI tasks. In addition, CESM is more than a coupling layer, it also includes a top level driver that supports the ability to call multiple components from the same MPI

tasks in a single executable but only to couple via the driver layer. Oasis3-MCT_3.0 does not have a driver layer and is driven by calls from inside the models. In practice, users could implement a top level driver using Oasis3-MCT_3.0, so Oasis3-MCT_3.0 can behave just like CESM plus it can behave in other ways. I am still a little unclear about whether the C-Coupler2 consists of a driver. If so, is it just a single executable system or does is support multiple executables? I believe none of the coupled systems discussed in this paragraph support multiple MPI tasks running on a single processor, and otherwise they are very similar in capabilities. The main difference is that CESM does not support coupling within a component compared to the other two. I think this paragraph should be clarified. It's difficult to read and the similarities and differences should be more clearly qualified.

page 22, paragraph beginning at line 22. It seems C-Coupler2 is using a file to coordinate MPI tasks between components. While this may be simpler than synchronizing with MPI, there is still the equivalent of a global barrier in the interaction. A component cannot know the tasks of other components until other components have written to the file. How does the C-Coupler2 ensure that other components have written to the file before the information is needed? What is "file" synchronization chosen over MPI?

page 34, line 30. The issues with 3D conservative coupling are the same as 2D. Even with areas, model areas and conservation method areas can differ and this needs to be taken into account with 2D conservative mapping. I do no believe there are any fundamental hurdles to extend 2D conservative coupling to 3D and there may be tools that already accomplish that.

Is the C-Coupler a hub coupler a component, is it just a layer in the system, is it the driver? I think C-Coupler1 was a hub and C-Coupler2 is a coupling layer, is that correct? It might be good to discuss this in the introduction and in regard to Figure 1.

With self-coupling or self-nesting on the same pes with the same executable and multiple grids, how does the C-Coupler2 address the issue of multi-instance data privacy

within the executable? It may not be enough just to instantiate a new domain or a new state. The underlying model has to meet specific and complex requirements to support that feature with regard to fully separating the memory of the two instances and most models do not. Does the C-Coupler2 actually support this and does it introduce any requirements on components to support that capability? For example, running multiple instances on concurrent pes does not create the same problems. Also, using the C-Coupler2 to couple internal data within a model that supports nesting is not difficult. It's not clear whether the C-Coupler2 supports self nesting on overlapping pes between a component model and another instance of the same component model. Section 4.6 suggests it can. How can that be? Maybe that could be clarified. This comes up in Section 3.6 and Section 4.6.

Does the C-Coupler2 support unstructured grids in 2D or 3D such as cubed sphere, non quadrilaterals, and other complex geometries? Does the on-line remapping support weight generation for those grids? Please indicate in the text.

The results show reasonable performance at moderately high resolution and pe counts. I think these results are adequate at this point, but it would be nice if there were an opportunity to test and publish results at higher resolution and higher task counts in the future, and I agree with the final statement on page 35, line 6.

I think section 4.1.1.1 to 4.1.1.8 could be removed and the user guide could be referenced instead. I think the API details are not needed in this paper. 4.1.1 could just a paragraph that provides a few sentences about the API and points to the user guide plus 4.1.1.9. That would be my recommendation, but will allow the authors to respond to this point.

Technical Comments:

Use of word "generations". Maybe it can be defined in first use as it is not clear or maybe another word is better, like "coupling interactions", "coupling procedures", or "coupling methods". I realize "coupling generations" is the output of the "coupling generator" but it's not the clearest language.

page 4, line 9, please define CoR1 better at the first instance and as needed in other locations in the paper and provide a reference if it exists.

page 5, line 14, chemistry can also be a separate package/component and 3D coupling in that case is important. I don't think you should say "is always included as an internal package".

page 5, line 28, please clarify that time varying is only supported in the vertical dimension, not the full 3D grid.

page 6, line 25, most couplers non block on sends and block on recvs to reduce deadlocking. Deadlocking is always an issue even for fully non-blocking communication. At some point, you have to block and check the data has been received before it's used.

page 10, line 13, locates is not a good word, try "a gridcell"

page 12, line 10, please clarify "model whose model coupling is fully served by other couplers but not C-Coupler2 is unnecessarily registered to C-Coupler2". Does this mean is should be registered, should not be registered, can be registered, or what? This sentence is unclear.

page 13, line 3, I find use of "timer" in this context to be confusing. I think you mean time, coupling period, alarm, coupling frequency or something similar. I understand timer is the word you have chosen to use in the interface, but it would be good to explain what "periodic timer" is in the context of the C-Coupler2. I think it defines the coupling period/frequency?

page 15, line 4, what is buf_mark?

page 23, line 23, please rewrite the first sentence in this paragraph. It is unclear.

[Figure]

2018.

---

## Short Comment (SC1) · 12 Mar 2018

Code that is to be made publicly available, must be made publicly available before the paper is accepted for GMD. The precise version of the code discussed in the manuscript must be made available. The current best practice is for this code to be uploaded to a public repository and a DOI assigned. The DOI should be cited in the manuscript. github/gitlab are inadequate because they do not readily link to the precise version of the code. However, making github code citable is not difficult; see: https://guides.github.com/activities/citable-code/

---

## Referee Comment (RC2) · Anonymous Referee #2 · 14 Mar 2018

General Comments:

This paper is a general overview of the C-Coupler2. The paper is largely a list of operations that the coupler performs and I the intended audience for this paper is potential users. Overall, I think the paper is fine, but it's too long and is not concise enough. For example, the motivation is mentioned multiple times and would be easier to understand if in one location. I suggest the authors re-think the motivation and re-tool the description section to focus on the main points. I provide specific suggestions below to make the paper shorter and more concise.

Specific Comments:

Page 1, line 23: Have couplers (as described here) been used in disciplines other

than environmental prediction? If not, combine these two sentences into one shorter sentence.

Page 1, line 29 to Page 2, line 15: There are two representative applications of C-Coupler1. What does representative mean in this context? How about just say the C-Coupler1 was built to support Chinese global and regional coupled modeling efforts? And then briefly mention limitations of C-Coupler1, which led to C-Coupler2. If the components models are mentioned, WRF, POM, MASNUM, and other abbreviations should be defined. Also, I am not sure FGOALS-g2 is a common known model and should be introduced.

Page 2 and 3, the list: Details of these features are disused elsewhere in this paper. They do not have to be discussed in the amount of detail here.

Page 3, line 25: This description of Figure 1 is not clear. Figure 1 may not be needed.

Page 4 through 8: I feel that section 3 can be summarized in a table similar to Table 10 (or use Table 10). The motivations are repeated in the Design section. Removing this section will increase readability for this paper.

Page 8, Line 20: The items in 1) have been mentioned before. Specifically, the text talking about the C-Coupler should be in the motivation and this section should be more about description.

Page 11, line 19ish: Throughout section 4.1, I was curious if there are defaults for each option.

Page 11, section 4.1.1.x: Is there a way to make the discussion of the APIs shorter? Could some of this be easily summarized in a table? And/or the API descriptions may be deleted.

Page 15, line19: coupling field instances from itself. I'm not exactly sure why a component would want field instances from itself? I may be missing something. Could you provide an example?

Page 15, line 35: How is a source fraction calculated? What are the multiple sources that would be used?

Page 15, line 40: Coupling procedures. Could you list some of these procedures at this point.

Page 17, line 4: CESM needs to be defined. Also, CESM has a lot of components, and I'm not sure what the model CESM means in this context.

Page 22, line 6: OASIS-MCT_3.0 needs an introduction.

Page 33, line 13: 960 cores seems small to stop the diagnostics. Many high resolution models require more than 960 cores.

Page 34, line 24: Guarantee is a strong word and you may not want backwards compatibility for all applications.

---

## Author Comment (AC1) · 2 May 2018

We thank Reviewer #1 for the comments and suggestions. We will modify the manuscript according to them in the revision stage. In the following, we will reply them one by one.

1. I would encourage the authors to publish additional results in the future detailing the performance cost of higher resolution and higher core counts tests and sharing the performance of 3D weight generation and coupling.

Response: Although it is difficult for us to find more processor cores to evaluate the performance cost of higher resolution and higher core counts tests, we will try to add the performance cost of dynamic 3-D weight generation and coupling in the revised

manuscript.

2. page23, lines25-29. The ability to run with lags properly is critical. Lags almost always create additional requirements on restart files and the ability to restart a model exactly (bit-for-bit) with lagged coupling fields should be a requirement if lags are going to be supported. It sounds like this is not currently supported in C-Coupler2? Maybe rather than saying "We therefore propose", it would be clearer to say something like "Lags are not fully supported in the current version, but in the future, the C-Coupler2 will ..."

Response: It is true that the capability of flexible lags setting will introduce significant technical challenges to achieve exact (bit-for-bit) restart at any case. In the latest code version of C-Coupler2 which will be publicly and formally released before the end of this month, these technical challenges have been fully resolved through a new feature of C-Coupler2: adaptive restart capability. In other words, C-Coupler2 can conveniently achieve exact restart no matter the setting of coupling lags, without any additional requirements. We will detail the implementation of the adaptive restart capability in the revised manuscript.

3. page 26, Section 4.5. Is the only reason MPI_Put and MPI_Get is used is to avoid possible exhaustion of MPI buffer space? That should be very rare in practice. Are there other reasons? Performance, ease of implementation, etc? Based on the description on Section 4.5, the MPI_Put/Get implementation sounds slower and more complicated than well managed MPI_ISend/IRecv implementations with MPI_Wait implemented appropriately. Are the authors happy with this implementation? Section 5.3 answers this question in part, but it might be nice to add a few more words in either section 4.5 or 5.3. I think one-side communication potentially helps with both MPI buffer usage and ability to have greater flexibility in coupling lags, but does not improve performance? How about implementation complexity?

Response: Originally, we thought that the MPI_Put/MPI_Get implementation was the

unique solution for achieving flexibility in coupling lags setting. Although it will not obviously increase the performance cost compared to the MPI_ISend/IRecv implementation in most cases, we encountered a significantly slow down case on a IBM system. After adding an MPI_ISend/IRecv implementation into C-Coupler2, we find that the MPI_ISend/IRecv implementation can also achieve flexibility in coupling lags. Therefore, in the latest C-Coupler2, there are both an MPI_Put/MPI_Get implementation and an MPI_ISend/IRecv implementation, where the latter one is used as default. Users are proposed to try the MPI_Put/MPI_Get implementation if an unpredictable deadlock is encountered. The complexity of the MPI_Put/MPI_Get implementation is much higher. The manuscript and user manual will be modified accordingly.

4. page 28, line 8. This is a nice feature. One has to be concerned about memory usage but this provides a nice way to allow extra flexibility in lags compared to other implementations.

Response: Extra memory usage and higher cost in achieving exact restart capability are unavoidable when coupling lags are big. We will remind this point in the revised manuscript.

5. page 22, paragraph beginning at line 6. I believe the comparison between Oasis3MCT_3.0, CESM, and C-Coupler2 is not particularly clear. The authors compare how components interact in different systems, but the definition of the component is not the same in each system. In CESM and the C-Coupler2, the component is defined by separation of scientific models. In Oasis, the component is defined by the separation of MPI tasks. In addition, CESM is more than a coupling layer, it also includes a top level driver that supports the ability to call multiple components from the same MPI tasks in a single executable but only to couple via the driver layer. Oasis3-MCT_3.0 does not have a driver layer and is driven by calls from inside the models. In practice, users could implement a top level driver using Oasis3-MCT_3.0, so Oasis3-MCT_3.0 can behave just like CESM plus it can behave in other ways. I am still a little unclear about whether the C-Coupler2 consists of a driver. If so, is it just

a single executable system or does is support multiple executables? I believe none of the coupled systems discussed in this paragraph support multiple MPI tasks running on a single processor, and otherwise they are very similar in capabilities. The main difference is that CESM does not support coupling within a component compared to the other two. I think this paragraph should be clarified. It's difficult to read and the similarities and differences should be more clearly qualified.

Response: Similar to Oasis3-MCT_3.0, C-Coupler2 also does not have a driver layer and is driven by calls from inside the models. C-Coupler2 has a definition of component model similar to CESM but different from Oasis3-MCT_3.0. C-Coupler2 improves its debugging capability based on its definition of component model. As a grid, a parallel decomposition or a field instance must be across all processes of the corresponding component model, C-Coupler2 can check whether all processes of a component model call the corresponding API at the same time, when registering a grid, a parallel decomposition or a field instance. We will improve this paragraph in the revised manuscript.

6. page 22, paragraph beginning at line 22. It seems C-Coupler2 is using a file to coordinate MPI tasks between components. While this may be simpler than synchronizing with MPI, there is still the equivalent of a global barrier in the interaction. A component cannot know the tasks of other components until other components have written to the file. How does the C-Coupler2 ensure that other components have written to the file before the information is needed? What is "file" synchronization chosen over MPI?

Response: Given that the coupled model covers 1000 MPI processes, while a component model A covers No. 0∼99 MPI processes and a component model B covers No. 900∼999 MPI processes, the coupling generation between A and B as well as the corresponding "file" synchronization only introduce a partial barrier among process 0∼99 and 900∼999, but not a global barrier. A (or B) will wait until the "file" of B (or A) is ready. We will clarify this point in the revised manuscript.

7. page 34, line 30. The issues with 3D conservative coupling are the same as 2D. Even with areas, model areas and conservation method areas can differ and this needs to be taken into account with 2D conservative mapping. I do not believe there are any fundamental hurdles to extend 2D conservative coupling to 3D and there may be tools that already accomplish that.

Response: In fact, we still do not have a good idea for achieving 3-D conservative coupling.

8. Is the C-Coupler a hub coupler a component, is it just a layer in the system, is it the driver? I think C-Coupler1 was a hub and C-Coupler2 is a coupling layer, is that correct? It might be good to discuss this in the introduction and in regard to Figure 1.

Response: Both C-Coupler1 and C-Coupler2 are libraries but not a hub coupler component. This will be stated clearly in the revised manuscript.

9. With self-coupling or self-nesting on the same pes with the same executable and multiple grids, how does the C-Coupler2 address the issue of multi-instance data privacy within the executable? It may not be enough just to instantiate a new domain or a new state. The underlying model has to meet specific and complex requirements to support that feature with regard to fully separating the memory of the two instances and most models do not. Does the C-Coupler2 actually support this and does it introduce any requirements on components to support that capability? For example, running multiple instances on concurrent pes does not create the same problems. Also, using the Coupler2 to couple internal data within a model that supports nesting is not difficult. It's not clear whether the C-Coupler2 supports self nesting on overlapping pes between a component model and another instance of the same component model. Section 4.6 suggests it can. How can that be? Maybe that could be clarified. This comes up in Section 3.6 and Section 4.6.

Response: C-Coupler2 does not allocate memory space for component models. Therefore, the issue of multi-instance data privacy within the same executable must

be addressed by the model itself. However, C-Coupler2 can easily identify different instances of the same field (the field names are the same) that are from different grid domains of the same component model or from different component models with the same type, so as to facilitate the implementation of model coupling. For example, given a self-nesting atmosphere model A with three levels of grid domains A1=>A2=>A3, and a elf-nesting ocean model O with the corresponding three levels of grid domains O1=>O2=>O3, C-Coupler2 can easily achieve atmosphere-ocean coupling at each grid level through the corresponding partial coupling generation (a coupling generation between A3 and O3, a coupling generation between A2 and O2, and a coupling generation between A1 and O1), while the configuration files for different partial coupling generations can be almost the same and very simple. In other words, C-Coupler2 can also facilitate the implementation of nesting of a coupled model. We will make clarification in the revised manuscript.

10. Does the C-Coupler2 support unstructured grids in 2D or 3D such as cubed sphere, non quadrilaterals, and other complex geometries? Does the on-line remapping support weight generation for those grids? Please indicate in the text.

Response: As C-Coupler2 still uses the remapping software CoR1 for remapping weight generation, the on-line remapping weight generation supports unstructured 2-D grids. We will state that in the revised manuscript.

11. The results show reasonable performance at moderately high resolution and pecounts. I think these results are adequate at this point, but it would be nice if there were an opportunity to test and publish results at higher resolution and higher task counts in the future, and I agree with the final statement on page 35, line 6.

Response: We only used moderately high resolution and pecounts in this paper mainly due to limited computing resource. Currently we are afraid of that we may not be able to successfully apply more computing resource.

12. I think section 4.1.1.1 to 4.1.1.8 could be removed and the user guide could be

referenced instead. I think the API details are not needed in this paper. 4.1.1 could just a paragraph that provides a few sentences about the API and points to the user guide plus 4.1.1.9. That would be my recommendation, but will allow the authors to respond to this point.

Response: We describe the APIs with details in order to introduce our consideration in how to design the APIs. We will try to shrink the content of section 4.1.1.1.

13. Technical Comments

Response: We will modify the manuscript according to the technical comments.

---

## Author Comment (AC2) · 2 May 2018

We thank Reviewer #2 for the comments and suggestions. We will modify the manuscript according to them in the revision stage. In the following, we will reply them one by one.

1. Page 1, line 23: Have couplers (as described here) been used in disciplines other than environmental prediction? If not, combine these two sentences into one shorter sentence.

Response: We will improve the manuscript accordingly in the revision.

2. Page 1, line 29 to Page 2, line 15: There are two representative applications of CCoupler1. What does representative mean in this context? How about just say the

[Figure]

CCoupler1 was built to support Chinese global and regional coupled modeling efforts? And then briefly mention limitations of C-Coupler1, which led to C-Coupler2. If the components models are mentioned, WRF, POM, MASNUM, and other abbreviations should be defined. Also, I am not sure FGOALS-g2 is a common known model and should be introduced.

Response: We will improve the manuscript accordingly in the revision.

3. Page 2 and 3, the list: Details of these features are disused elsewhere in this paper. They do not have to be discussed in the amount of detail here.

Response: We will try to shrink the content in this part in the revision.

4. Page 3, line 25: This description of Figure 1 is not clear. Figure 1 may not be needed.

Response: We will try to improve the description of Figure 1.

5. Page 4 through 8: I feel that section 3 can be summarized in a table similar to Table 10 (or use Table 10). The motivations are repeated in the Design section. Removing this section will increase readability for this paper.

Response: We will try to shrink the description of the motivations in the revision.

6. Page 8, Line 20: The items in 1) have been mentioned before. Specifically, the text talking about the C-Coupler should be in the motivation and this section should be more about description.

Response: We will improve the manuscript accordingly.

7. Page11, line 19ish: Throughout section 4.1, I was curious if there are defaults for each option.

Response: We will try to introduce more clearly about default options.

8. Page 15, line19: coupling field instances from itself. I'm not exactly sure why

a component would want field instances from itself? I may be missing something. Could you provide an example?

Response: We will add an example about this aspect.

9. Page 15, line 35: How is a source fraction calculated? What are the multiple sources that would be used?

Response: The source fraction used by C-Coupler2 is given by the component model via the input parameter of the corresponding API. We will introduce this aspect more clearly.

10. Page 15, line 40: Coupling procedures. Could you list some of these procedures at this point.

Response: We will introduce this aspect more clearly.

11. Page 17, line 4: CESM needs to be defined. Also, CESM has a lot of components, and I'm not sure what the model CESM means in this context.

Response: We will improve the manuscript accordingly.

12. Page 22, line 6: OASIS-MCT_3.0 needs an introduction.

Response: We will improve the manuscript accordingly.

13. Page 33, line 13: 960 cores seems small to stop the diagnostics. Many high resolution models require more than 960 cores.

Response: We only used moderately high resolution and pecounts in this paper mainly due to limited computing resource. Currently we are afraid of that we may not be able to successfully apply more computing resource although we will try.

14. Page 34, line 24: Guarantee is a strong word and you may not want backwards compatibility for all applications.

Response: We will improve the manuscript accordingly.

---

## Author Comment (AC3) · 2 May 2018

Dear Chief-executive editor,

Thanks a lot for your suggestions.

We are now preparing the final public release of the C-Coupler2 code. The C-Coupler2 code will be pulicly released before the end of this month. We will not only give a DOI of the C-Coupler2 code following your suggestions, but also show how to get further updates of the C-Coupler2 code in the revised manuscript.

Best regards,

Li

---

## Author Response (AR1)

**Part 1: Responses to the Chief Editor.**

We thank the Chief Editor for the comments and suggestions.

1. Code that is to be made publicly available, must be made publicly available before the paper is accepted for GMD. The precise version of the code discussed in the manuscript must be made available. The current best practice is for this code to be uploaded to a public repository and a DOI assigned. The DOI should be cited in the manuscript. github/gitlab are inadequate because they do not readily link to the precise version of the code. However, making github code citable is not difficult; see: https://guides.github.com/activities/citable-code/

Response: The source code of C-Coupler2 is publicly available. Please refer to the code availability section.

**Part 2: Responses to the Reviewer #1**

We thank Reviewer #1 for the comments and suggestions. We have modified the manuscript accordingly. In the following, we will reply them one by one.

1. I would encourage the authors to publish additional results in the future detailing the performance cost of higher resolution and higher core counts tests and sharing the performance of 3D weight generation and coupling.

Response: We are sorry that we fail to obtain more processor cores for evaluating the performance cost under higher resolution and higher core counts. However, we share the results and performance of 3D coupling in Section 5.5 and Figure 15-17, in the revised manuscript.

2. page23, lines25-29. The ability to run with lags properly is critical. Lags almost always create additional requirements on restart files and the ability to restart a model exactly (bit-for-bit) with lagged coupling fields should be a requirement if lags are going to be supported. It sounds like this is not currently supported in C-Coupler2? Maybe rather than saying "We therefore propose", it would be clearer to say something like "Lags are not fully supported in the current version, but in the future, the C-Coupler2 will ..."

Response: A new feature, adaptive restart capability, has been implemented in C-Couper2, for conveniently achieving exact restart of coupling fields no matter the setting of coupling lags and no matter the implementation of coupling. Please refer to Section 4.8 (P32L1~P35L17) of the revised manuscript.

3. page 26, Section 4.5. Is the only reason MPI_Put and MPI_Get is used is to avoid possible exhaustion of MPI buffer space? That should be very rare in practice. Are there other reasons? Performance, ease of implementation, etc? Based on the description on Section 4.5, the MPI_Put/Get implementation sounds slower and more complicated than well managed MPI_ISend/IRecv implementations with MPI_Wait implemented appropriately. Are the authors happy with this implementation? Section 5.3 answers this question in part, but it might be nice to add a few more words in either section 4.5 or 5.3. I think one-side communication potentially helps with both MPI buffer usage and ability to have greater flexibility in coupling lags, but does not improve performance? How about implementation complexity?

Response: In the released code version of C-Coupler2, non-blocking data transfer has two implementations. The first is the default option based on two-sided MPI communication, while the second is an additional option based on one-sided MPI communication. Please refer to section 4.5 (P27L11~P27L28, and P29L4~P29L13) of the revised manuscript.

4. page 28, line 8. This is a nice feature. One has to be concerned about memory usage but this provides a nice way to allow extra flexibility in lags compared to other implementations.

Response: The extra memory usage has been reminded in the revised manuscript (please refer to P29L11).

5. page 22, paragraph beginning at line 6. I believe the comparison between Oasis3MCT_3.0, CESM, and C-Coupler2 is not particularly clear. The authors compare how components interact in different systems, but the definition of the component is not the same in each system. In CESM and the C-Coupler2, the component is defined by separation of scientific models. In Oasis, the component is defined by the separation of MPI tasks. In addition, CESM is more than a coupling layer, it also includes a top level driver that supports the ability to call multiple components from the same MPI tasks in a single executable but only to couple via the driver layer. Oasis3-MCT_3.0 does not have a driver layer and is driven by calls from inside the models. In practice, users could implement a top level driver using Oasis3-MCT_3.0, so Oasis3-MCT_3.0 can behave just like CESM plus it can behave in other ways. I am still a little unclear about whether the C-Coupler2 consists of a driver. If so, is it just a single executable system or does is support multiple executables? I believe none of the coupled systems discussed in this paragraph support multiple MPI tasks running on a single processor, and otherwise they are very similar in capabilities. The main difference is that CESM does not support coupling within a component compared to the other two. I think this paragraph should be clarified. It's difficult to read and the similarities and differences should be more clearly qualified.

Response: The corresponding paragraph has been modified accordingly. Please refer to P22L26~P23L27 in the revised manuscript.

6. page 22, paragraph beginning at line 22. It seems C-Coupler2 is using a file to coordinate MPI tasks between components. While this may be simpler than synchronizing with MPI, there is still the equivalent of a global barrier in the interaction. A component cannot know the tasks of other components until other components have written to the file. How does the C-Coupler2 ensure that other

components have written to the file before the information is needed? What is "file" synchronization chosen over MPI?

Response: When a component model wants know the MPI processes of any other component model, all its MPI processes will wait to read the corresponding XML file. This point has been clarified in the revised manuscript (P23L23).

7. page 34, line 30. The issues with 3D conservative coupling are the same as 2D. Even with areas, model areas and conservation method areas can differ and this needs to be taken into account with 2D conservative mapping. I do not believe there are any fundamental hurdles to extend 2D conservative coupling to 3D and there may be tools that already accomplish that.

Response: In fact, we still do not have a good idea for achieving 3-D conservative coupling, so we do not modify the corresponding context. If required, we can remove some discussion in future revision.

8. Is the C-Coupler a hub coupler a component, is it just a layer in the system, is it the driver? I think C-Coupler1 was a hub and C-Coupler2 is a coupling layer, is that correct? It might be good to discuss this in the introduction and in regard to Figure 1.

Response: Both C-Coupler1 and C-Coupler2 are libraries but not a hub coupler component. Please refer to P3L30 and P23L6 of the revised manuscript.

9. With self-coupling or self-nesting on the same pes with the same executable and multiple grids, how does the C-Coupler2 address the issue of multi-instance data privacy within the executable? It may not be enough just to instantiate a new domain or a new state. The underlying model has to meet specific and complex requirements to support that feature with regard to fully separating the memory of the two instances and most models do not. Does the C-Coupler2 actually support this and does it introduce any requirements on components to support that capability? For example, running multiple instances on concurrent pes does not create the same problems. Also, using the Coupler2 to couple internal data within a model that supports nesting is not difficult. It's not clear whether the C-Coupler2 supports self nesting on overlapping pes between a component model and another instance of the same component model. Section 4.6 suggests it can. How can that be? Maybe that could be clarified. This comes up in Section 3.6 and Section 4.6.

Response: This point has been clarified in the revised manuscript. Please refer to P30L27~P30L31 in the revised manuscript.

10. Does the C-Coupler2 support unstructured grids in 2D or 3D such as cubed sphere, non quadrilaterals, and other complex geometries? Does the on-line remapping support weight generation for those grids? Please indicate in the text.

Response: Unstructured grids are supported in the on-line remapping weight generation. Please refer to P20L28 in the revised manuscript.

11. The results show reasonable performance at moderately high resolution and pecounts. I think these results are adequate at this point, but it would be nice if there were an opportunity to test and publish results at higher resolution and higher task counts in the future, and I agree with the final statement on page 35, line 6.

Response: We only used moderately high resolution and pecounts in this paper mainly due to limited computing resource. We are sorry that we failed to obtain more processor cores for testing and evaluation because most super computers in China are full of usage. We will keep to apply more processor cores in the future.

12. I think section 4.1.1.1 to 4.1.1.8 could be removed and the user guide could be referenced instead. I think the API details are not needed in this paper. 4.1.1 could just a paragraph that provides a few sentences about the API and points to the user guide plus 4.1.1.9. That would be my recommendation, but will allow the authors to respond to this point.

Response: In the revised manuscript, we shrink Section 4.1.1 where most of tables are removed. As flexible APIs is an important feature of C-Coupler2 and most of scientists do not know C-Coupler, we hope we can keep some introduction to the C-Coupler2 APIs with the remaining content that briefly introduce the motivations or considerations for the APIs.

13. Use of word "generations". Maybe it can be defined in first use as it is not clear or maybe another word is better, like "coupling interactions", "coupling procedures", or "coupling methods". I realize "coupling generations" is the output of the "coupling generator" but it's not the clearest language.

Response: "Coupling generation" has been modified into "coupling procedure generation" throughout the revised manuscript. "Coupling procedure generation" means generating coupling procedures. Please refer to P2L33 to P3L4.

14. page 4, line 9, please define CoR1 better at the first instance and as needed in

other locations in the paper and provide a reference if it exists.

Response: Please refer to P2L28 in the revised manuscript.

15. page5,line14,chemistry can also be a separate package/component and 3D coupling in that case is important. I don't think you should say "is always included as an internal package".

Response: The manuscript has been modified accordingly. Please refer to P5L19 to P5L20.

16. page 5, line 28, please clarify that time varying is only supported in the vertical dimension, not the full 3D grid.

Response: The manuscript has been modified accordingly. Please refer to P6L2.

17. page 6, line 25, most couplers non block on sends and block on recvs to reduce deadlocking. Deadlocking is always an issue even for fully non-blocking communication. At some point, you have to block and check the data has been received before it's used.

Response: The manuscript has been modified accordingly. Please refer to Section 3.5 (P6L15 to P6L28).

18. page 10, line 13, locates is not a good word, try "a gridcell"

Response: The manuscript has been modified accordingly. Please refer to P10L26.

19. page 12, line 10, please clarify "model whose model coupling is fully served by other couplers but not C-Coupler2 is unnecessarily registered to C-Coupler2". Does this mean is should be registered, should not be registered, can be registered, or what? This sentence is unclear.

Response: The manuscript has been modified accordingly. Please refer to P12L21.

20. page 13, line 3, I find use of "timer" in this context to be confusing. I think you mean time, coupling period, alarm, coupling frequency or something similar. I understand timer is the word you have chosen to use in the interface, but it would

be good to explain what "periodic timer" is in the context of the C-Coupler2. I think it defines the coupling period/frequency?

Response: The manuscript has been modified accordingly. Please refer to P13L18

21. page 15, line 4, what is buf_mark?

Response: It is a non-negative integer mark given by users, employed in each coupling field instance to separate multiple coupling field instances in the same component model, on the same grid, and on the same parallel decomposition. Please refer to P15L23 in the revised manuscript.

22. page 23, line 23, please rewrite the first sentence in this paragraph. It is unclear.

Response: This sentence has been rewritten. Please refer to P24L15 in the revised manuscript.

**Part 3: Responses to the Reviewer #2**

We thank Reviewer #2 for the comments and suggestions. We have modified the manuscript accordingly. In the following, we will reply them one by one.

1. Page 1, line 23: Have couplers (as described here) been used in disciplines other than environmental prediction? If not, combine these two sentences into one shorter sentence.

Response: The manuscript has been modified accordingly. Please refer to P1L23 and P1L24.

2. Page 1, line 29 to Page 2, line 15: There are two representative applications of CCoupler1. What does representative mean in this context? How about just say the CCoupler1 was built to support Chinese global and regional coupled modeling efforts? And then briefly mention limitations of C-Coupler1, which led to C-Coupler2. If the components models are mentioned, WRF, POM, MASNUM, and other abbreviations should be defined. Also, I am not sure FGOALS-g2 is a common known model and should be introduced.

Response: The manuscript has been modified accordingly. Please refer to P1L28 to P2L10.

3. Page 2 and 3, the list: Details of these features are disused elsewhere in this paper. They do not have to be discussed in the amount of detail here.

Response: This part can be viewed as a summary for each new feature of C-Coupler2. We really hope this part can be retained to make readers get to know all features of C-Coupler2 at the beginning and then decide whether to read the details of this paper.

4. Page 3, line 25: This description of Figure 1 is not clear. Figure 1 may not be needed.

Response: The description of Figure 1 has been improved in the revised manuscript. Please refer to P3L29 to P3L3.

5. Page 4 through 8: I feel that section 3 can be summarized in a table similar to

Table 10 (or use Table 10). The motivations are repeated in the Design section. Removing this section will increase readability for this paper.

Response: As the motivations for the new features in C-Coupler2, we think that Section 3 is necessary to make C-Coupler2 as well as its software structure in Fig.5 more understandable, and should be retained. Section 3 and Section 4 have been modified accordingly, to avoid the repetition between them.

6. Page 8, Line 20: The items in 1) have been mentioned before. Specifically, the text talking about the C-Coupler should be in the motivation and this section should be more about description.

Response: It seems that this part which focuses on the difference between C-Coupler1 and C-Coupler2 on software structures has not been introduced before, which is related to the detailed implementation of C-Coupler2. Section 3 is about the motivations for each new feature of C-Coupler2. So we still propose to retain this part in Section 4.

7. Page11, line 19ish: Throughout section 4.1, I was curious if there are defaults for each option.

Response: The default values of some parameters of APIs have been introduced in the user guide.

8. Page 15, line19: coupling field instances from itself. I'm not exactly sure why a component would want field instances from itself? I may be missing something. Could you provide an example?

Response: For example, C-Coupler2 can achieve coupling between the physical package and the dynamic core in the same component model. Please refer to P16L9.

9. Page 15, line 35: How is a source fraction calculated? What are the multiple sources that would be used?

Response: The source fraction used by C-Coupler2 is given by the component model via the input parameter of the corresponding API. It is required for conservative remapping. Please refer to P16L16 in the revised manuscript.

10. Page 15, line 40: Coupling procedures. Could you list some of these procedures at

this point.

Response: A coupling procedure can include a set of operations such as data transfer, data interpolation, data type transformation, and data averaging when necessary. Please refer to P3L1 to P3L3 in the revised manuscript.

11. Page 17, line 4: CESM needs to be defined. Also, CESM has a lot of components, and I'm not sure what the model CESM means in this context.

Response: Here CESM means the main driver of CESM that is shared by all component models. Please refer to P17L27 in the revised manuscript.

12. Page 22, line 6: OASIS-MCT_3.0 needs an introduction.

Response: OASIS3-MCT_3.0 is the latest coupler in OASIS family. The manuscript is modified accordingly. Please refer to P22L26.

13. Page 33, line 13: 960 cores seems small to stop the diagnostics. Many high resolution models require more than 960 cores.

Response: We only used moderately high resolution and pecounts in this paper mainly due to limited computing resource. We are sorry that we failed to obtain more processor cores for testing and evaluation because most super computers in China are full of usage. We will keep to apply more processor cores in the future.

14. Page 34, line 24: Guarantee is a strong word and you may not want backwards compatibility for all applications.

Response: The manuscript has been modified accordingly. Please refer to P39L13.

Part 3: a marked-up manuscript version

[revised manuscript text omitted]

[revised manuscript text omitted]

带格式式表格

[Figure]

Figure 11Figure 12  Initialization cost for coupling two toy models with C-Coupler2 on a supercomputer with Intel Xeon CPUs and an InfiniBand network.

[Figure]

Figure 12Figure 13  Comparison of data transfer times (for 100 ping-pong couplings) between a one-sided and a two-sided implementation, with the same configuration as Fig. 11Fig. 12.

[Figure]

 Memory use of C-Coupler2 for the toy coupled model considered in Section 5.2.

[Figure]

(a) Temperature (T)

[Figure]

(a) Zonal wind speed (U)

Figure 15  The temperature (a) and zonal wind speed (U) from GAMIL2 to GEOS-Chem (GC) at the 500 hPa level at two different model time.

[Figure]

(a) Temperature (T)

[Figure]

(b) Zonal wind speed

Figure 16  The global vertical profile of the temperature (a) and zonal wind speed (U) from GAMIL2 to GEOS-Chem (GC) at the 500 hPa level at two different model time.

[Figure]

Figure 17  The parallel speedup of dynamic 3-D coupling (for 100 ping-pong couplings) between the two component models, with a new configuration derived from the configuration used in Fig. 12. The speedup is normalized to the time at 15 cores per component model (1583 s).

---

## Editor Decision (ED1)

Dear Author,

Thank you for this revised version of your manuscript. I think it has considerably improved compared to the first version but there are still many important points that need clarification. In particular, I consider that in many cases, you don't reply to the reviewer's comments in a satisfactory way. Therefore I suggest you analyse and reply to my following comments before the manuscript can be considered for publication. My comments are classified into 4 lists: first the comments related to Reviewer 1 comments, than the one related to Reviewer 2 comments, then my own additional major comments, and finally my additional minor comments.

Reviewer 1 comments:

**2 – Details on lags and restarts**

In the revised manuscript, there is a whole new section on adaptive restart capability. Here are some comments on this issue and new section:

- P24, l20-24: I don't understand these lines. You propose to "…to extend the simulation period to guarantee correct simulation of the model states in the concerned simulation period", but then you add a whole new section 4.8 on "Adaptive restart capability" describing how C-Coupler2 supports restart capability even with lags (fig 11 a and 11 c). Also, in the last paragraph of this new section, you conclude that "C-Coupler2 currently does not guarantee exact restart capability under such kind of coupling lag specification". Please clarify what is supported or not by the C-Coupler2 in terms of restart capability with lags.
- I don't understand the first paragraph, p 32, l6-12. I think that in your text you suppose that Comp1 and Comp2 are stopped after 600s and restarted; is it the case? If so, it is not clear, either in your text, or on Fig 11a. Please clarify. If this is the case, then I suppose that when Comp1 and Comp2 are restarted:
  - Comp1 exports exp1_1 at 1200s (that will be received by Comp2 at 1800s) and then imports imp1_1 that is exported as exp2_1 by Comp2 at 1200s.
  - Comp2 exports exp2_1 and then tries to import imp2_1. Comp1 produced the corresponding exp1_1 during the previous run at 600s; therefore it has to be in the restart file. And then the simulation can go on.
  So I conclude that the only coupling field that has to be in the restart file is exp1_1 from Comp1 at 600s and I don't understand why you write: "… at the second and third iterations", p 32, l12. Please correct or clarify.
- Regarding case 11c, I understand, as you write (P33, l1), that it is unnecessary for comp1 to export exp1_1 at 1200s, but then I don't understand where imp2_1, needed by Comp2 at 600s, will come from? Please clarify.
- P33, l11: what do you mean with "almost" in "almost at each time step"? Is it or is it not required at each time step?
- P33, last line, P34, l1-3. You write "When a receiver component … the values of the import field instance will be read from the corresponding NetCDF restart data file"; will this be done automatically or is it that the model has to explicitly call a *CCPL_start_restart_read_IO* ?

- P34, l4-21, fig 11 (a): the example is too difficult to follow without a graph illustrating the different steps. If you want to keep the example, please add an explanatory graph.

**3 - One-sided vs two-sided communication: your updated section 4.5 is fine for me but for the following details:**
- P29, L9: What do you mean by "when all existing message passing buffers are unavailable"; please rephrase.

**5 - Comparison with OASIS3-MCT and CESM.**

I am sorry to write that this comparison does not seem much clearer to me in the revised manuscript. I would advice to completely remove this comparison and just insist on C-Coupler2 characteristics.
For example, does C-Coupler2 allow coupling between two components that run sequentially on the same MPI tasks? Please specify.
I think you should keep the first part of the sentence "Similar with OASIS3-MCT_3.0, C-Coupler2 also works as a library without a driver layer and is driven by calls from the models", but please remove "or coupling layers such as CESM" as I think this does not apply to C-Coupler2 and this last part of the sentence is confusing.
Please remove sentences like "Therefore, each component model of CESM can be treated as a component model of C-Coupler2, and coupling between different component models of CESM can still be treated as coupling between different component models. " or "Therefore, CESM can be treated as a unique component model of OASIS3-MCT_3.0, while any coupling between CESM component models can be treated as a coupling between different grids of OASIS3-MCT_3.0." as I think that these are more confusing than informative.

**6 - Using a file to coordinate MPI tasks between components**

I consider that you don't answer the reviewer's comments. In particular, you did not add anything regarding the fact that even if file synchronisation may be simpler than synchronizing with MPI, there is still the equivalent of a global barrier in the interaction. Also you did not answer the question on how does the C-Coupler2 ensure that all components have written to the file before any other component gets the information. Please clarify these two points.

**7 - 3D conservative coupling**

Again, I consider that you did not answer the reviewer's comment. Given his/her remarks, I think that what you write is not correct. There is no specific problem extending 2D conservative remapping based on areas to 3D conservative remapping based on volumes. The problem you mention "a component model … may have its own specific way to diagnose 3-D conservation" is also present in 2D. Please review this section.

**9 – Self nesting on overlapping pes**

I am sorry to say that it is still not clear to me if C-Coupler2 supports self nesting between a component model and another instance of the same component model running on the same or overlapping MPI tasks. Please clarify. Sorry if I missed something.

**17 – Deadlocking and non-blocking communication**

You write that you made modifications in the text to address the reviewer's comment on the fact that, even with non-blocking communication, you have at one point to check if the data has been received, but I don't see any related modifications in the text.

Reviewer 2 comments:

**4 - Figure 1**

I agree that Figure 1 is not needed; it does not help understanding the text, which itself is fine.

**7 - Default for options**

Could you mention somewhere in the text that the default for each option is described in the User Guide?

**9 – Source fractions**

It is still not clear to me what "source fractions" are and how they are involved in the conservative remapping. Can you clarify?

**10 – Coupling procedure**

I agree with reviewer that the list, or at least examples, should be repeated at this point. Or please add a reference to the list provided p3, l2.

**11 – CESM driver**

For clarity, maybe replace "… such as CESM consist …" by "… , such as the CESM driver , consist …"

**12 – OASIS3-MCT**

For clarity, please replace "Note that the latest coupler OASIS3-MCT_3.0 …" by "Note that the latest version of the OASIS coupler, OASIS3-MCT_3.0 …"

**14 – "guarantee"**

I think that replacing "guarantee" by "do our best" is not appropriate. Either you target backward compatibility and you do what is needed for this, either you don't target backward compatibility. See also my "Additional major comment" on Section 6.

Additional major comments

**Section 4.1.1.7 - Two designs for the coupling procedure generation**

What do you mean by "Two designs … are compared."
Do you mean that they were compared and that you decided to implement only the second design in C-Coupler2? If so, why do you insist so much (3 paragraphs) on the first design?
Do you mean that the two designs are implemented in C-Coupler2 and that you compare here in this paper the two? (I don't think this is the case).
Please clarify.

**p22, lines 6 & 7: please clarify what is effectively done in practice i.e. using the average value or the instantaneous field at its last activation. Is it the user who chooses what to do through the configuration file?**

**Section 6**

I think this section is particularly week and should be reviewed. This section is not a general "Discussion and conclusion", it is just a (not very precise) description of some future plans for C-Coupler2.
First I am not sure what "integrating an external coupling algorithms" precisely means. In this context, I think you mean integrating some external calculation/transformation routines? If so using "coupling algorithm" is too vague.
Second, see my remark above (#14 – guarantee) about the sentence "However, we will try our best …"
Finally, the discussion on 3D conservative remapping needs to be reviewed (see also my remark above "#7 - 3D conservative coupling").

Additional minor comments

- P1, l28-29: It could be nicer to write "Two coupled models were built using C-Coupler1" instead of "There are two coupled models with C-Coupler1".
- P3, l5: Replace "in default …" by "by default"
- P6, l5: Replace "Coupling generator" by "Coupling procedure generation"
- P6, l17-18: Stating "In a coupled model, a component model always executes both data send and receive operations (i.e., two-way coupled)" is wrong. There are one-way coupled models into which one model performs only data send and one model performs only data receive. Maybe modify the sentence like this: "In a two-way coupled model, a component model always executes both data send and receive operations."
- P12, l22: Replace "… unnecessary register a model …" by "… unnecessary to register a model …"
- P24, l15: Replace "… by an receiver …" by "… by a receiver …"
- P29, l23-24: Please rewrite sentence "For a regional model without self-nesting capability (i.e., it can only manage a unique grid domain), C-Coupler2 can it help achieve self-nesting capability as follows." Maybe "C-Coupler2 can also help achieve self-nesting in a regional model that does not originally support this possibility." would be better?

- P30, l27: Please rewrite "Similarly, it can also benefit from C-Coupler2 to nest a regional model into a different model.". Maybe "Similarly, C-Coupler2 can be used to nest a regional model into a different model." would be better?
- P33, l14-15: change "(Users can disable … the API. Please refer … details)," for "(users can disable … the API; please refer … details),"

---

## Author Response (AR2)

Dear Editor,

Thanks a lot for reviewing our manuscript and for your comments and suggestions that essentially help us to improve the manuscript.

In this revised manuscript, we add a new author, Guangwen Yang (ygw@tsinghua.edu.cn), because he gave essential contribution for revising the manuscript, especially in evaluating the dynamic 3-D coupling capability. I am sorry that we forgot to add him in the last revised manuscript. We are looking forward your understanding and agreement for this application. Thanks a lot.

Next, we'd like to reply the comments and suggestions one by one.

1. P24, l20-24: I don't understand these lines. You propose to "…to extend the simulation period to guarantee correct simulation of the model states in the concerned simulation period", but then you add a whole new section 4.8 on "Adaptive restart capability" describing how C-Coupler2 supports restart capability even with lags (fig 11 a and 11 c). Also, in the last paragraph of this new section, you conclude that "C-Coupler2 currently does not guarantee exact restart capability under such kind of coupling lag specification". Please clarify what is supported or not by the C-Coupler2 in terms of restart capability with lags.

Response: In the revised manuscript, the paragraph from P24 L16 to L32 has been modified with a new example to illustrate why the stop time of the simulation run should be extended in some cases with coupling lags. This part is not contradictory with the adaptive restart capability. Following the example in this part, if the simulation stop time is still 86400 s, it will fail to restart the simulation from the model time of 86400 s in a restart run as the restart data files are not complete, and C-Coupler2 will automatically select the previous restart write time for a "*continue*" run. For the question what is supported or not by the C-Coupler2 in terms of restart capability with lags, we add a new paragraph (P36 L3-L8) for summarizing the adaptive restart capability.

2. About Section 4.8

1) I don't understand the first paragraph, p 32, l6-12. I think that in your text you suppose that Comp1 and Comp2 are stopped after 600s and restarted; is it the case? If so, it is not clear, either in your text, or on Fig 11a. Please clarify. If this is the case, then I suppose that when Comp1 and Comp2 are restarted:

   a) Comp1 exports exp1_1 at 1200s (that will be received by Comp2 at 1800s) and then imports imp1_1 that is exported as exp2_1 by Comp2 at 1200s.

   b) Comp2 exports exp2_1 and then tries to import imp2_1. Comp1 produced the corresponding exp1_1 during the previous run at 600s; therefore it has to be in the restart file. And then the simulation can go on.

   So I conclude that the only coupling field that has to be in the restart file is exp1_1 from Comp1 at 600s and I don't understand why you write: "… at the second and third iterations", p 32, l12. Please correct or clarify.

2) Regarding case 11c, I understand, as you write (P33, l1), that it is unnecessary for comp1 to export exp1_1 at 1200s, but then I don't understand where imp2_1, needed by Comp2 at 600s, will come from? Please clarify.

3) P33, l11: what do you mean with "almost" in "almost at each time step"? Is it or is it not required at each time step?

4) P33, last line, P34, l1-3. You write "When a receiver component … the values of the import field instance will be read from the corresponding NetCDF restart data file"; will this be done automatically or is it that the model has to explicitly call a CCPL_start_restart_read_IO ?

5) P34, l4-21, fig 11 (a): the example is too difficult to follow without a graph illustrating the different steps. If you want to keep the example, please add an explanatory graph.

Response: Section 4.8, especially the examples, have been significantly modified according to the above comments, and Figure 10 has been corrected and improved with additional 5 graphs. Please refer to the context from P32L17 to P33L18, from P34L21 to P35L5.

3. #3 - One-sided vs two-sided communication: your updated section 4.5 is fine for me but for the following details:

1) P29, L9: What do you mean by "when all existing message passing buffers are unavailable"; please rephrase.

Response: This sentence has been modified as "C-Coupler2 checks each import interface and will adaptively increase the message passing buffers when required" (P29L23).

4. #5 - Comparison with OASIS3-MCT and CESM.

I am sorry to write that this comparison does not seem much clearer to me in the revised manuscript. I would advice to completely remove this comparison and just insist on C-Coupler2 characteristics.

For example, does C-Coupler2 allow coupling between two components that run sequentially on the same MPI tasks? Please specify.

I think you should keep the first part of the sentence "Similar with OASIS3-MCT_3.0, C-Coupler2 also works as a library without a driver layer and is driven by calls from the models", but please remove "or coupling layers such as CESM" as I think this does not apply to C-Coupler2 and this last part of the sentence is confusing.

Please remove sentences like "Therefore, each component model of CESM can be treated as a component model of C-Coupler2, and coupling between different component

models of CESM can still be treated as coupling between different component models. " or "Therefore, CESM can be treated as a unique component model of OASIS3-MCT_3.0, while any coupling between CESM component models can be treated as a coupling between different grids of OASIS3-MCT_3.0." as I think that these are more confusing than informative.

Response: Thanks a lot for the suggestions. This comparison has been removed from the context. Please refer to Section 4.2.

5. #6 - Using a file to coordinate MPI tasks between components

I consider that you don't answer the reviewer's comments. In particular, you did not add anything regarding the fact that even if file synchronisation may be simpler than synchronizing with MPI, there is still the equivalent of a global barrier in the interaction. Also you did not answer the question on how does the C-Coupler2 ensure that all

components have written to the file before any other component gets the information. Please clarify these two points.

Response: We think that file synchronization may be the only choice to avoid global synchronization in a partial coupling procedure generation, because the component models involved in the same coupling procedure generation may not know the MPI processes (for example, the ID of the process in the global communicator MPI_COMM_WORLD) of each other. The first and second paragraphs in Section 4.3 (P22L28-P23L28) has been modified for clarification.

6. #7 - 3D conservative coupling

Again, I consider that you did not answer the reviewer's comment. Given his/her remarks, I think that what you write is not correct. There is no specific problem extending 2D conservative remapping based on areas to 3D conservative remapping based on volumes. The problem you mention "a component model … may have its own specific way to diagnose 3-D conservation" is also present in 2D. Please review this section.

Response: We have removed this part in Section 6 in the revised manuscript.

7. #9 - Self nesting on overlapping pes

I am sorry to say that it is still not clear to me if C-Coupler2 supports self nesting between a component model and another instance of the same component model running on the same or overlapping MPI tasks. Please clarify. Sorry if I missed something.

Response: C-Coupler2 does not support self nesting between a component model and another instance of the same component model running on the same or overlapping MPI tasks. This point has been clarified at P30L6-L7.

8. #17 - Deadlocking and non-blocking communication

You write that you made modifications in the text to address the reviewer's comment on the fact that, even with non-blocking communication, you have at one point to check if the data has been received, but I don't see any related modifications in the text.

Response: Before the import interface copies out coupling field values from its message passing buffer, it first checks whether new coupling field values have been received.

Besides coupling field values, the export interface will send a model time tag to the import interface at the same time. Given an MPI process of the import interface, if all model time tags in its message passing buffer (different model time tags correspond to different MPI processes of the export interface) are the same and later than the tags of the last receive, it means that new coupling field values have been received. This point has been introduced in the revised manuscript. Please refer to P28L11 to L27.

9. #4 - Figure 1

I agree that Figure 1 is not needed; it does not help understanding the text, which itself is fine.

Response: Figure 1 has been removed and the first paragraph in Section 2 (P3L29 to P4L11) has be modified accordingly.

**7 - Default for options**

Could you mention somewhere in the text that the default for each option is described in the User Guide?

Response: Section 4.10 is newly added for introducing some default options. Please refer to P37L2-L30.

10. #9 – Source fractions

It is still not clear to me what "source fractions" are and how they are involved in the conservative remapping. Can you clarify?

Response: Fraction means the area fraction of atmosphere, ocean, land surface or sea ice in each cell of the source grid. It will be the input when registering a fraction-based remap interface for conservative remapping. Please refer to P16L15-L20 in the revised manuscript.

11. #10 – Coupling procedure

I agree with reviewer that the list, or at least examples, should be repeated at this point. Or please add a reference to the list provided p3, l2.

Response: The list is introduced from P25L14 to L29. A reference "(please refer to Section 4.3 for details)" has been added at P3L3.

12. #11 – CESM driver

For clarity, maybe replace "… such as CESM consist …" by "… , such as the CESM driver , consist …"

Response: The context has been modified accordingly. Please refer to P17L28.

13. #12 – OASIS3-MCT

For clarity, please replace "Note that the latest coupler OASIS3-MCT_3.0 …" by "Note that the latest version of the OASIS coupler, OASIS3-MCT_3.0 …"

Response: The corresponding paragraph has been removed.

14. #14 – "guarantee"

I think that replacing "guarantee" by "do our best" is not appropriate. Either you target backward compatibility and you do what is needed for this, either you don't target backward compatibility. See also my "Additional major comment" on Section 6.

Response: This statement has been modified as "However, keeping backwards compatibility will be a primary goal for future C-Coupler versions". Please refer to P41L11.

15. #Section 4.1.1.7 - Two designs for the coupling procedure generation

What do you mean by "Two designs … are compared."

Do you mean that they were compared and that you decided to implement only the second design in C-Coupler2? If so, why do you insist so much (3 paragraphs) on the first design?

Do you mean that the two designs are implemented in C-Coupler2 and that you compare here in this paper the two? (I don't think this is the case).

Please clarify.

Response: The development of C-Coupler2 has experienced two the both design and the second design can be viewed as an upgrade of the first design. This part has been modified accordingly. Please refer to P17L21-L28, P18L4.

16. # p22, lines 6 & 7: please clarify what is effectively done in practice i.e. using the average value or the instantaneous field at its last activation. Is it the user who chooses what to do through the configuration file?

Response: Using the average value or the instantaneous field is specified by users when registering an import interface. This point has been clarified. Please refer to P25L24-L26.

17. # Section 6

I think this section is particularly week and should be reviewed. This section is not a general "Discussion and conclusion", it is just a (not very precise) description of some future plans for C-Coupler2.

First I am not sure what "integrating an external coupling algorithms" precisely means. In this context, I think you mean integrating some external calculation/transformation routines? If so using "coupling algorithm" is too vague.

Second, see my remark above (#14 – guarantee) about the sentence "However, we will try our best …"

Finally, the discussion on 3D conservative remapping needs to be reviewed (see also my remark above "#7 - 3D conservative coupling").

Response: The title of Section 6 has been modified as "future work", the discussion about 3-D conservation has been removed, and "external coupling algorithm" has been replaced by "external calculation routine" throughout the whole manuscript. Please refer to Section 6 (P40L28-P41L14).

18. Additional minor comments

Response: Thanks a lot. The manuscript has been improved according to the suggestions.

[revised manuscript text omitted]
 (corresponding to the red words of ,*"do restart write"* in the second iteration in Fig. 10(a)), in a restart run of the coupled model restarted from the  model time of 600 s (Fig. 10(b)), after the both component models read in the corresponding restart data (corresponding to the red words of *"do restart read"* in the second iteration in Fig. 10(b)) and next advance the model time, *comp2* will enter the third iteration with the model time of 1200 s and the coupling interface *imp2_3* will import the coupling field instance values exported by the coupling interface *exp1_2* of *comp1* at its model time of 600 s. However, *comp1* will also enter the third iteration with the model time of 1200 s and will never execute *exp1_2* again. Therefore, besides the values imported by *imp2_2* (these values may be used by *comp2* before executing *imp2_3*), the values imported by *imp2_3* should also be included in the restart data files corresponding to the model time of 600 s . This example indicates that the restart data files corresponding to a restart write model time should include the coupling field instance values at different model time corresponding to a positive lag on a coupling connection.

We'd like to set the second example based on the coupled model setting in Fig. 10(c). There are no coupling lags on the coupling connections between the two component models, and the only difference between the two component models is that, they have different orders for writing restart data files and advancing model time in each iteration of the main loop. Given that the whole coupled model should prepare restart data corresponding to the model time of 600 s, the

5   component model *comp1* should  produce restart data files at the second iteration, while *comp2* should  produce restart data files at the first iteration because its model time has already been advanced to 600 s. In a restart run of the coupled model restarted from the model time of 600 s (Fig. 10(d)), after the both component models read in the corresponding restart data (corresponding to the red words of *"do restart read"* in the second iteration of *comp1* and in the first iteration of *comp2* in Fig. 10(d)),

10  *comp2* will enter the second iteration with the model time of 600 s and  *imp2_2* will import the coupling field instance values exported by the coupling interface *exp1_2* of *comp1* at its model time of 600 s. However, *comp1* will enter the third iteration with the model time of 1200 s and will never execute *exp1_2* again. Therefore, besides the values imported by *imp2_1* (these values may be used by *comp2* before executing *imp2_2*), the values imported by *imp2_2* should also be included in the restart data files

15  corresponding to the model time of 600 s.  This example indicates that the restart data files corresponding to a restart write model time may need to include the coupling field instance values at different model time even when there is no lag on any coupling connection.

As shown in Fig. 10(d), *comp2* will execute the coupling interface *exp2_2* after the coupled model run is restarted.

20  *exp2_2* will try to export the coupling field instance values to the coupling interface *imp1_2* of *comp1*, however, *imp1_2* will never be executed again in the restart run. Therefore, the data export by *exp2_2* should be bypassed, to avoid deadlocks. Similarly, regarding the third example corresponding to Fig. 10(e) with a coupling lag of -600 s from *comp1* to *comp2*, in the restart run corresponding to Fig. 10(f), the data export by the coupling interface *exp1_3* of *comp1* should also be bypassed. These examples indicate that it may need to bypass the data export of some export interfaces at some model time after

25  restarting the coupled model run.

[revised manuscript text omitted]

Fig. 10(g) shows  an example about the above implementation based on the coupled model

5 setting in Fig. 10(a) , where the red operations are restart write related. Given that the unique restart timer will be on at the model time of 600 s, for each of *comp1* and *comp2,* the API *CCPL_do_restart_write_IO* will write the coupling field instance values imported   at the model time of 600 s (corresponding to *imp1_2* and *imp2_2*) into the corresponding NetCDF restart data file and mark 600 s as the restart writing model time. At the third iteration with the model time of 1200 s, *imp2_3* of comp2 will obtain the coupling field instance values that are exported by *exp1_2* of *comp1* at its model time of 600 s. As 600 s is the same with but not later than the restart writing model time, according to the above step 4,

10 *imp2_3* will write its obtained  coupling field instance values  *comp2*  into the corresponding NetCDF restart data file. According to the above step 5, *comp1* will produce the corresponding binary restart data file when advancing the model time at the third iteration , while *comp2* will produce the corresponding binary restart data file when advancing the model time at the fourth iteration . Fig. 10(h) shows a restart run of the coupled model restarted from the model time of 600

[revised manuscript text omitted]

**4.10  Default options for using C-Coupler2**

Many C-Coupler2 APIs have optional input parameters. Moreover, some configuration files are optional. Default options will be used when optional input parameters or configuration files are not specified or provided. Here we'd like to list out some default options for using C-Coupler2 as follows. For more default options, please refer to the user guide.

1) The API *CCPL_register_component* for registering a component model has an optional input parameter "*considered_in_ancestor_coupling_gen*". Its default value is *true*, which means the component model registered currently will be involved in the family coupling procedure generation of its parent component model. For example, given the component models in Fig. 5 and that "*considered_in_ancestor_coupling_gen*" is not specified when registering *comp1* and *comp2* while it has been set to *false* when registering *comp3*, *comp1* and *comp2* will be involved in the global coupling procedure generation while *comp3* will be not.

2) The API *CCPL_define_single_timer* for defining a periodic timer has an optional input parameter "*remote_lag_count*".

Its default value is *0*, which means there will be no coupling lag on the corresponding coupling connections.

3) The API *CCPL_register_H2D_grid_via_global_data* and *CCPL_register_H2D_grid_via_local_data* have an optional input parameter "*mask*" that specifies whether each grid cell is active or not. Its default value is *1*, which means all grid cells are active.

4) The API *CCPL_register_import_interface* for registering an import interface has an optional input parameter "*necessity*". It is an array, each element of which specifies whether the corresponding import field instance is necessary (with value of 1) or optional (with value of 0). Its default value is *1*, which means all import field instances are necessary. When executing an import interface, if a necessary import field instance has not been connected (the provider has not been found and the corresponding coupling procedures have not been generated), the whole model run will be stopped with an error report.

5) Coupling connection configuration files are optional (in some cases when no coupling connection configuration file has been given, component models can still be successfully coupled together) and will become necessary when multiple providers for a coupling field instance in an import interface have been detected in a coupling procedure generation.

3)6)Remapping configuration files are optional and the default remapping configuration will be used when no remapping configuration file is provided. In the default remapping configuration, the bilinear remapping algorithm is used for remapping the "*state*" fields between horizontal grids, the conservative remapping algorithm is used for remapping the "*flux*" fields between horizontal grids, the linear remapping algorithm is used for remapping in both the vertical dimension and the time dimension, and the remapping weights between different grids will be generated by C-Coupler2.

[revised manuscript text omitted]

[Figure]

(c) An initial run of a two-way coupled model without coupling lags.

[Figure]

(d) A restart run of the coupled model in Figure (c) restarted from the model time of 600 s

(e) Two-way coupling with a lag of 600 s from *comp1* to *comp2*

[Figure]

(e) An initial run of a two-way coupled model with a lag of -600 s from *comp1* to *comp2*

[Figure]

(f) A restart run of the coupled model in Figure (e) restarted from the model time of 600 s

(g) An initial run corresponding to Figure (a). The first iteration is not shown. The red operations are restart write related

(h) A restart run of the coupled model in Figure (g) restarted from the model time of 600 s. The red operations are restart read related

Figure 11Figure 10  Sample restart requirements under -different coupling lag settings. The grey words indicate that the corresponding operations will be bypassed or not executed.

[Figure]

Figure 11  Initialization cost for coupling two toy models with C-Coupler2 on a supercomputer with Intel Xeon CPUs and an InfiniBand network.

[Figure]

Figure 13Figure 12  Comparison of data transfer times (for 100 ping-pong couplings) between a one-sided and a two-sided implementation, with the same configuration as Fig. 12Fig. 11.

[Figure]

Figure 14Figure 13  Memory use of C-Coupler2 for the toy coupled model considered in Section 5.2.

[Figure]

(a) Temperature (T)

[Figure]

5                                                (a) Zonal wind speed (U)

Figure 14  The temperature (a) and zonal wind speed (U) from GAMIL2 to GEOS-Chem (GC) at the 500 hPa level

at two different model time.

[Figure]

(a) Temperature (T)

[Figure]

5                             (b) Zonal wind speed

Figure 16Figure 15  The global vertical profile of the temperature (a) and zonal wind speed (U) from GAMIL2 to GEOS-Chem (GC) at the 500 hPa level at two different model time.

[Figure]

Figure 17Figure 16 The parallel speedup of dynamic 3-D coupling (for 100 ping-pong couplings) between the two component models, with a new configuration derived from the configuration used in Fig. 12Fig. 11. The speedup is normalized to the time at 15 cores per component model (1583 s).

---

## Author Response (AR3)

Dear Editor,

Thanks a lot for giving us a series of suggestions that are essentially important for further improving the manuscript. Next, we'd like to reply them one by one.

1. P.2, L.14-16: In "Considering this, and that the next C-Coupler versions may not be compatible with C-Coupler1, we did not aim to achieve wide usage of C-Coupler1, but sought to overcome the limitations while guaranteeing backwards compatibility in subsequent C-Coupler versions", the first part of the sentence "the next C-Coupler versions may not be compatible with C-Coupler1" seems contradictory to the last part of the sentence "while guaranteeing backwards compatibility in subsequent C-Coupler versions". Please modify and clarify.

Response: It seems unnecessary to state about the backwards compatibility here that will be discussed in the Section of summary and future work. The corresponding context therefore has been changed to "With such a limitation, C-Coupler1 did not achieve wide usage". Please refer to P2L13 of the revised manuscript.

2. P.6, L.7-8: you should not write "Most existing couplers require the user to develop explicitly all coupling procedures"; some do but I would not write that most of them do.

Response: This sentence has been changed into "Some existing couplers require the user to develop explicitly all coupling procedures". Please refer to P6L6 of the revised manuscript.

3. If I understand well, the software structure illustrated on Figure 1 was never implemented as such. C-Coupler1 and C-Coupler-2 structures are illustrated on Figure 2 and Figure 4 respectively. Therefore, what is represented on Figure 1 should be referred in the caption and in the text to as "General software design of C-Coupler" (i.e. with "design" instead of "structure"). Please change Figure 1 captions and the text referring to Figure 1 accordingly. For example, I think that

writing on P.9, L.2-3 "This software structure is similar to that of the original C-Coupler software design (Fig. 1)" and on P.9, L.4 "The original design of the C-Coupler …" would be clearer.

Response: The manuscript has been improved accordingly. Please refer to P4L2, P4L10, P6L12, P9L3, P9L4, P9L9, and the title of Figure 1 (P44) in the revised manuscript.

4. P.17, L.21: after "… has experienced two designs", please add (as explained in your reply) "The second design can be viewed as an upgrade of the first design and is currently implemented in C-Coupler2".

Response: This sentence has been added into the revised manuscript. Please refer to P17L21~L22.

5. P.19, L.1: This whole section 4.1.1.9 is too detailed and related Figures 6 and 7 are too hard to read. I suggest moving them to a supplement and referencing to the supplement at the beginning of section 4.1.1.

Response: Section 4.1.1.9 and the related Figures 6 and 7 have been moved to the supplement and are referenced in the revised manuscript. Please refer to P12L4~L5.

6. P.24, L.8: please add "The choice is done by the user when registering the import interface" there as the explanation on P.25, L.24-26 is too far below.

Response: This sentence has been added to the revised manuscript. Please refer to P23L12~L13.

7. P.24, L.15: for clarity, please add "based on the user's choice".

Response: These words have been added to the revised manuscript. Please refer to P23L16.

8. Section 4.3, P.22-26: should be subdivided into subsections for better readability, I suggest:

    o "4.3.1 Creation of MPI communicators" : from P.22,L.28 to P.23,L.28

o "4.3.2 Timers matching and lags": from P.23,L.29 to P.24,L.32

o "4.3.3 Steps for coupling procedure generation": from P.24,L.33 to P.26, L.2

Response: The above subtitles have been inserted into the revised manuscript. Please refer to P22L2, P23L1 and P24L7.

9. P.27, L.22-30: I am not sure why this paragraph is written putting emphasis on OASIS3-MCT and not on C-Coupler2. The current paper is on C-Coupler2 and not on OASIS3-MCT so this should be modified. Describe how non-blocking data transfer is implemented in C-Coupler2 and then maybe conclude "Similar non-blocking data transfer is implemented in OASIS3-MCT". Also sentence starting with "To achieve non-blocking data transfer …" and ending with "… last put of the same coupling field" is way to cumbersome; please rephrase and simplify.

Response: The corresponding context has been modified in the revised manuscript. Please refer to P27L2~L8.

10. P.28, L.17-19: You write "if all model time tags in its message passing buffer … are the same and later than the tags of the last receive, it means that new coupling field values have been received"; I don't understand how a new coupling field can be associated with same tags than the last receive; I would say that a new coupling field is only associated with tags later than the tag of the last receive. Please clarify or correct if I am right.

Response: The corresponding context has been modified in the revised manuscript. Please refer to P27L27~L28.

11. P.29,L.6: I still don't understand, as in my first round of review, what do you mean by "it is unnecessary to execute imp2_1"; I have the same question with imp2_1 and imp2_2 on Fig 9(c). The component model needs those input fields to run, where will it get it from ???

Response: The imp2_1 will not be executed, because it corresponds to exp1 executed at the model time of -600 s that is earlier than the start time of the model run, while the

coupling field values imported by imp2 can be initialized via data files or coupling in the initialization stage of the coupled model. The corresponding context (P28L16~L18) and the title of Figure 7 has been modified in the revised manuscript.

12. P.30,L.1: change "unbreakable" for "unsolvable"; I understand that 9(e) case is unsolvable even with one-sided MPI communication where message passing buffers can be increased adaptively; if I am right, please state it in the text by adding "(even with one-sided MPI communication)" after "deadlock".

Response: "unbreakable" has been changed to "unsolvable" at P28L31 and P29L12 of the revised manuscript. "even with one-sided MPI communication" has also been added (P29L12).

13. P.30,L31-32: how do you define "coupling latencies"?

Response: "coupling latencies" means coupling lags. All "coupling latencies" has been changed into "coupling lags" throughout the revised manuscript.

14. Section "4.8 Adaptive restart capability": I consider that this section is still too complex and very hard to follow and understand. I propose to move this whole section and Figure 10 to a supplement and to provide in the paper just a summary of the C-Coupler2 adaptive restart capability. (Note that on p.34, "bybrid" should be "hybrid").

Response: We are sorry that Section 4.8 is still very hard to follow and understand. As Section 4 introduces the design and implementation of C-Coupler2, in the revised manuscript, we list out the implementation of the adaptive restart capability in Section 4.8, in addition to the summary, while the examples as well as Figure 10 have been moved into the supplement. We are sorry that the current revision does not strictly follow the above suggestion. We will further modify the manuscript if such modification is still improper.

15. Section 4.9 is too detailed for a paper; this section looks more like a section of the User Guide. In my comment from the previous round of review, I did not ask for a whole new section on defaults, I just asked to mention somewhere that default for each option is described in the User Guide. Please remove section 4.9 and just add somewhere in the text that default for each option is described in the User Guide.

Response: Section 4.9 has been removed. Most default options have been introduced in the User Guide. The User Guide will be further improved according to this manuscript.

16. P.40, L.1: how can you speculate about the C-Coupler2 memory usage based on the assessment of OASIS3-MCT? This seems quite unjustified to me. Please clarify or modify.

Response: We are sorry about such speculation. Section 5.4 has been modified accordingly. Please refer to P36L13~L19 of the revised manuscript.

17. Section "6 Future work": this section was improved compared to the previous version but I think it is still too weak. I think you should turn this section into "Summary and future work" including a first half page summarizing the differences between C-Coupler1 and C-Coupler2, i.e. highlighting the most important points from Table 2 and then going on with what you currently have in your section "6 Future work" with one paragraph on the integration of external calculation routines (current P.41, L.1-9), one paragraph on the coupling configuration interface backward compatibility (current P.41, L.10-11) and one paragraph on initialization cost and memory use (current P.41, L.12-14)

Response: The title and the organization of Section 6 has been modified according to the above suggestions. Please refer to P37L13~P38L16 of the revised manuscript.

18. Suggestion for English improvement

Response: Thanks a lot for these suggestions. The manuscript has been modified accordingly. Please refer to P1L15~L16, P1L19, P1L27, P1L27~L28, P3L12, P3L29~L30, P6L18, P6L20, P8L19, P9L15~L17, P21L19, P23L2, P23L16~L17,

P24L1, P26L7, P27L32, P28L32, P29L12, P28L12, P29L8, and P37L14 of the revised manuscript.

[revised manuscript text omitted]
 (corresponding to the red words of *"do restart write"* in the second iteration in Fig. 10(a)), in a restart run of the coupled model restarted from the model time of 600 s (Fig. 10(b)), after the both component models read in the corresponding restart data (corresponding to the red words of *"do restart read"* in the second iteration in Fig. 10(b)) and next advance the model time, *comp2* will enter the third iteration with the model time of 1200 s and the coupling interface *imp2_3* will import the coupling field instance values exported by the coupling interface *exp1_2* of *comp1* at its model time of 600 s. However, *comp1* will also enter the third iteration with the model time of 1200 s and will never execute *exp1_2* again. Therefore, besides the values imported by *imp2_2* (these values may be used by *comp2* before executing *imp2_3*), the values imported by *imp2_3* should also be included in the restart data files corresponding to the model time of 600 s. Thiss~~ 1~3  indicate that the restart data files corresponding to a restart write model time  may need to include the coupling field instance values at different model time corresponding to a positive  lag on a coupling connection or an order between writing restart data files and advancing model time,

~~We'd like to set the second example based on the coupled model setting in Fig. 10(c). There are no coupling lags on the coupling connections between the two component models, and the only difference between the two component models is that, they have different orders for writing restart data files and advancing model time in each iteration of the main loop. Given that the whole coupled model should prepare restart data corresponding to the model time of 600 s, the component model *comp1* should produce restart data files at the second iteration, while *comp2* should produce restart data files at the first iteration because its model time has already been advanced to 600 s. In a restart run of the coupled model restarted from the model time of 600 s (Fig. 10(d)), after the both component models read in the corresponding restart data (corresponding to the red words of *"do restart read"* in the second iteration of *comp1* and in the first iteration of *comp2* in Fig. 10(d)), *comp2* will enter the second iteration with the model time of 600 s and it coupling interface *imp2_2* will import the coupling~~

field instance values exported by the coupling interface *exp1_2* of *comp1* at its model time of 600 s. However, *comp1* will enter the third iteration with the model time of 1200 s and will never execute *exp1_2* again. Therefore, besides the values imported by *imp2_1* (these values may be used by *comp2* before executing *imp2_2*), the values imported by *imp2_2* should also be included in the restart data files corresponding to the model time of 600 s. This example indicates that the restart data files corresponding to a restart write model time may need to include the coupling field instance values at different model time even when there is no lag on any coupling connection.

As shown in Fig. 10(d), *comp2* will execute the coupling interface *exp2_2* after the coupled model run is restarted. *exp2_2* will try to export the coupling field instance values to the coupling interface *imp1_2* of *comp1*, however, *imp1_2* will never be executed again in the restart run. Therefore, the data export by *exp2_2* should be bypassed, to avoid deadlocks. Similarly, regarding the third example corresponding to Fig. 10(e) with a coupling lag of -600 s from *comp1* to *comp2*, in the restart run corresponding to Fig. 10(f), the data export by the coupling interface *exp1_3* of *comp1* should also be bypassed. These examples indicate thatand that it may need to bypass the data export of some export interfaces at some model time after restarting the coupled model run.

To conveniently achieve exact restart for coupling fields under any setting of coupling lags and any order between writing restart data files and advancing model time, the restart manager of C-Coupler2 provides an adaptive restart capability implemented as follows (For further illustration about the implementation, please refer to Example 4 in Section 2 of the Supplement).

- **The restart manager conducts restart writing as follows:**

[revised manuscript text omitted]

each of *comp1* and *comp2*, the API *CCPL_do_restart_write_IO* will write the coupling field instance values imported at the model time of 600 s (corresponding to *imp1_2* and *imp2_2*) into the corresponding NetCDF restart data file and mark 600 s as the restart writing model time. At the third iteration with the model time of 1200 s, *imp2_3* of comp2 will obtain the coupling field instance values that are exported by *exp1_2* of *comp1* at its model time of 600 s. As 600 s is the

5 same with but not later than the restart writing model time, according to the above step 4, *imp2_3* will write its obtained coupling field instance values into the corresponding NetCDF restart data file. According to the above step 5, *comp1* will produce the corresponding binary restart data file when advancing the model time at the third iteration, while *comp2* will produce the corresponding binary restart data file when advancing the model time at the fourth iteration. Fig. 10(h) shows a restart run of the coupled model restarted from the model time of 600 s, where red operations are restart read

[revised manuscript text omitted]

**5.0   Default options for using C-Coupler2**

Many C-Coupler2 APIs have optional input parameters. Moreover, some configuration files are optional. Default options will be used when optional input parameters or configuration files are not specified or provided. Here we'd like to list out some default options for using C-Coupler2 as follows. For more default options, please refer to the user guide.

0)   The API *CCPL_register_component* for registering a component model has an optional input parameter "*considered_in_ancestor_coupling_gen*". Its default value is *true*, which means the component model registered currently will be involved in the family coupling procedure generation of its parent component model. For example, given the component models in Fig. 5 and that "*considered_in_ancestor_coupling_gen*" is not specified when registering *comp1* and *comp2* while it has been set to *false* when registering *comp3*, *comp1* and *comp2* will be involved in the global coupling procedure generation while *comp3* will be not.

0)   The API *CCPL_define_single_timer* for defining a periodic timer has an optional input parameter "*remote_lag_count*". Its default value is *0*, which means there will be no coupling lag on the corresponding coupling connections.

0)   The API *CCPL_register_H2D_grid_via_global_data* and *CCPL_register_H2D_grid_via_local_data* have an optional input parameter "*mask*" that specifies whether each grid cell is active or not. Its default value is *1*, which means all grid cells are active.

0)   The API *CCPL_register_import_interface* for registering an import interface has an optional input parameter "*necessity*". It is an array, each element of which specifies whether the corresponding import field instance is necessary

(with value of 1) or optional (with value of 0). Its default value is *1*, which means all import field instances are necessary. When executing an import interface, if a necessary import field instance has not been connected (the provider has not been found and the corresponding coupling procedures have not been generated), the whole model run will be stopped with an error report.

5    11) Coupling connection configuration files are optional (in some cases when no coupling connection configuration file has been given, component models can still be successfully coupled together) and will become necessary when multiple providers for a coupling field instance in an import interface have been detected in a coupling procedure generation.

12) Remapping configuration files are optional and the default remapping configuration will be used when no remapping configuration file is provided. In the default remapping configuration, the bilinear remapping algorithm is used for
10    remapping the "*state*" fields between horizontal grids, the conservative remapping algorithm is used for remapping the "*flux*" fields between horizontal grids, the linear remapping algorithm is used for remapping in both the vertical dimension and the time dimension, and the remapping weights between different grids will be generated by C-Coupler2.

[revised manuscript text omitted]

(c) An initial run of a two-way coupled model without coupling lags.

(d) A restart run of the coupled model in Figure (c) restarted from the model time of 600 s

(e) An initial run of a two-way coupled model with a lag of 600 s from *comp1* to *comp2*

(f) A restart run of the coupled model in Figure (e) restarted from the model time of 600 s

(g) An initial run corresponding to Figure (a). The first iteration is not shown. The red operations are restart write related

(h) A restart run of the coupled model in Figure (g) restarted from the model time of 600 s. The red operations are restart read related

Figure 10 Sample restart requirements under different coupling lag settings. The grey words indicate that the corresponding operations will be bypassed or not executed.

[Figure]

Figure 11Figure 8  Initialization cost for coupling two toy models with C-Coupler2 on a supercomputer with Intel Xeon CPUs and an InfiniBand network.

[Figure]

Figure 12Figure 9 Comparison of data transfer times (for 100 ping-pong couplings) between a one-sided and a two-sided implementation, with the same configuration as Fig. 11Fig. 8.

[Figure]

Figure 10  Memory use of C-Coupler2 for the toy coupled model considered in Section 5.2.

[Figure]

(a) Temperature (T)

[Figure]

5                                         (a) Zonal wind speed (U)

Figure 14Figure 11   The temperature (a) and zonal wind speed (U) from GAMIL2 to GEOS-Chem (GC) at the 500 hPa level at two different model time.

[Figure]

(a) Temperature (T)

[Figure]

(b) Zonal wind speed

Figure 12  The global vertical profile of the temperature (a) and zonal wind speed (U) from GAMIL2 to GEOS-Chem (GC) at the 500 hPa level at two different model time.

[Figure]

Figure 16Figure 13  The parallel speedup of dynamic 3-D coupling (for 100 ping-pong couplings) between the two component models, with a new configuration derived from the configuration used in Fig. 11Fig. 8. The speedup is normalized to the time at 15 cores per component model (1583 s).

---

## Author Response (AR4)

Dear Editor,

Thanks a lot for the new suggestions and your efforts in handling our manuscript.

The manuscript has been improved accordingly. Please refer to P27L2, P27L5-7, P36L13, P37L26-27, and P38L10, respectively. Moreover, we slightly modified the code availability section (P38L18-19), and added a new project number in the *acknowledgements* section (P38L25).

Regarding to default options, default values of optional parameters of each API (if have) have been introduced in the user guide, while there is no separate section of default options. The user guide will be improved accordingly.

With best regards,

Li

[revised manuscript text omitted]